# Stabilizing ultrasmall Au clusters for enhanced photoredox catalysis

Bo Weng[1,2], Kang-Qiang Lu[1,2], Zichao Tang[3], Hao Ming Chen [4] & Yi-Jun Xu [1,2]

Recently, loading ligand-protected gold (Au) clusters as visible light photosensitizers onto various supports for photoredox catalysis has attracted considerable attention. However, the efficient control of long-term photostability of Au clusters on the metal-support interface remains challenging. Herein, we report a simple and efficient method for enhancing the photostability of glutathione-protected Au clusters (Au GSH clusters) loaded on the surface of $SiO_2$ sphere by utilizing multifunctional branched poly-ethylenimine (BPEI) as a surface charge modifying, reducing and stabilizing agent. The sequential coating of thickness controlled $TiO_2$ shells can further significantly improve the photocatalytic efficiency, while such structurally designed core-shell $SiO_2$-Au GSH clusters-BPEI@$TiO_2$ composites maintain high photostability during longtime light illumination conditions. This joint strategy via interfacial modification and composition engineering provides a facile guideline for stabilizing ultrasmall Au clusters and rational design of Au clusters-based composites with improved activity toward targeting applications in photoredox catalysis.

[1] State Key Laboratory of Photocatalysis on Energy and Environment College of Chemistry, Fuzhou University, 350116 Fuzhou, People's Republic of China. [2] College of Chemistry, New Campus, Fuzhou University, 350116 Fuzhou, People's Republic of China. [3] State Key Laboratory of for Physical Chemistry of Solid Surfaces, Collaborative Innovation Center of Chemistry for Energy Materials, College of Chemistry and Chemical Engineering, Xiamen University, 361005 Xiamen, People's Republic of China. [4] Department of Chemistry, National Taiwan University, 106 Taipei, Taiwan. Correspondence and requests for materials should be addressed to H.M.C. (email: haomingchen@ntu.edu.tw) or to Y.-J.X. (email: yjxu@fzu.edu.cn)

Noble metal nanoparticles (MNPs) are uniquely suited for heterogeneous catalysis because of their relatively high specific surface areas and abundant active centers[1–5]. The size of MNPs has been shown to be one of the most important factors that dictates the performance of a catalyst[5]. However, the catalytically active nano-sized MNPs are at a thermodynamic unstable state, which are inclined to sinter—they form fewer, larger particles—under thermal and/or light irradiation conditions[2,6–10]. For instance, it has been demonstrated that nano-sized gold nanoparticles irradiated by a femtosecond laser-induced white-light supercontinuum can initiate a growth of larger nanoparticles[9]. Additionally, the shape of silver nano-spheres can be converted into triangular nanoprisms under the illumination of light at wavelengths between 350 and 700 nm[10]. Thus, a variety of solid support strategies have been utilized in practical catalytic systems for stabilization of the small MNPs in heterogeneous catalysis since the size and shape of MNPs provide important control over the catalytic efficiency[6,7].

Recently, the ultrasmall MNPs with size regime of 1–3 nm in diameter (often called metal clusters, e.g., gold (Au) clusters) that are composed of a specific number of metal atoms and of ligands have received considerable attention due to their distinctive physicochemical properties[11–16], which are different from their plasmonic counterpart owing to the quantum size effect, surface geometric effect (e.g., surface atom arrangement and low-coordinated atoms), and high surface-to-volume ratio[17,18]. These ultrasmall ligands (typically thiolate)-protected Au clusters exhibit discrete electronic structure[18,19], which have been regarded as one of the prototypes of highly tunable molecular-sized materials for catalysis and solar energy conversion. In particular, the thiolate-protected Au clusters, typically glutathione-protected Au clusters (Au GSH clusters), are able to serve as both photo-sensitizer and catalytic center for multifunctional use in photo-catalytic redox systems, such as water splitting, selective organic synthesis, and pollutants degradation[20–29]. However, such ultrasmall Au GSH clusters suffer from serious instability under light irradiation due to its extremely high surface energy and large surface[15]. The loading of ultrasmall Au GSH clusters onto different supports is often inefficient to avoid coalescence and agglomeration of these Au GSH clusters[15,27–30], which thereby results in either the difficulty of unambiguous photo-catalytic mechanism assignment or a loss of photocatalytic activity[15,27,28].

Regarding the Au clusters-based photocatalysts, seminal works have been reported for understanding the observed aggregation of Au GSH clusters at the semiconductor/metal interface. For instance, the irradiation of high-energy electron beams can lead to the aggregation of Au GSH clusters into larger Au NPs (~5 nm diameter) after 2 h continuous visible light illumination ($\lambda > 420$ nm) over Au GSH-sensitized Pt/TiO$_2$ photocatalysts[28]. It has also been reported that the photogenerated electrons from Au GSH clusters and/or TiO$_2$ can be captured by Au (I) component on the outer layer of Au GSH clusters (i.e., Au(0)@Au(I)-GSH) to produce Au NPs and the photocatalytic decomposition of GSH ligands tethered on the Au clusters surface can also lead to formation of metallic Au NPs[27]. Recently, a diffusion/aggregation mechanism has been proposed to elucidate the photoinduced coalescence of Au GSH clusters to larger metallic Au NPs at the Au/TiO$_2$ interface under light illumination[15]. Despite various studies on disclosing underlying mechanism of transformation of ultrasmall molecular-like Au GSH clusters to Au NPs, the effective control of Au GSH clusters with long-term stability on the substrates under in situ photo-irradiation conditions still remains a challenge, which becomes the main bottleneck for the development of Au clusters-based catalysts systems for photoredox applications[15,27,28].

Herein, we report a simple, combinatorial approach to stabilize ultrasmall Au GSH clusters on the SiO$_2$ sphere support and improve their photocatalytic efficiency by using multifunctional branched poly-ethylenimine (BPEI) for surface modification and coating thickness controlled semiconductor TiO$_2$ shell for interfacial composition engineering. The BPEI is shown to behave as a surface charge modifying, reducing and stabilizing agent for interfacial modification of SiO$_2$-Au GSH clusters composites. Specifically, the size and structure of Au GSH clusters can be well retained over 10 h under continuous visible light irradiation ($\lambda > 420$ nm) because of the high reductive ability of BPEI to inhibit the organic ligand oxidation process over Au GSH clusters. In addition, while simultaneously maintaining the photo-stability of Au GSH clusters, the sequential coating of TiO$_2$ shells for constructing core-shell SiO$_2$-Au GSH clusters-BPEI@TiO$_2$ (SABT) nanostructures has been demonstrated to significantly ameliorate the photoactivity by regulating the photoelectro-chemical and adsorption properties of SABT composites synergistically. Therefore, the strategy via interfacial modification and composition manipulation by coating semiconductor shell provides an efficient way for stabilizing Au clusters with improved photocatalytic activity, which is anticipated to enable the broad development of Au clusters-based composite system for photoredox applications in solar energy conversion.

## Results

**Structural characterizations of Au GSH clusters**. The successful fabrication of Au GSH clusters has been proved by transmission electron microscopy (TEM) analysis and the characteristic emissive properties of Au GSH clusters[31]. As apparently shown in Supplementary Fig. 1a, TEM image shows the uniform dispersed ultrasmall Au GSH clusters with mean diameter around 1.4 nm (Supplementary Fig. 1b). The Au GSH clusters solution exhibits yellow emissive properties under the blacklight illumination (Supplementary Fig. 1d and e), which is different from that of plasmonic Au counterparts[15,27,28]. UV-vis absorption spectrum of Au GSH clusters in Supplementary Fig. 1c suggests that the Au GSH clusters show an absorption onset at ~520 nm with a distinct shoulder around 400 nm, which is attributed to the highest occupied molecular orbital-lowest unoccupied molecular orbital (HOMO-LUMO) transition originated from the ligand-to-metal charge transfer[15], indicating that Au GSH clusters could be used as visible light photosensitizers. The photoluminescence (PL) excitation spectrum exhibits a maximum at 400 nm, which coincides well with the absorption shoulder observed in the absorption spectrum. The PL emissive spectra of Au GSH clusters with different excitation wavelengths in Supplementary Fig. 1f show a low energy emission band with the peak maximum at 605 nm, which is ascribed to the triplet metal-centered state[32], and the shape of the PL spectra is independent of the excitation wavelength[15]. The large Stokes shift in the emission, where the absorption band shoulder appears at around 400 nm and the emission maximum is seen at 605 nm, is consistent with the excited state being a ligand-metal charge transfer type[15,33]. In addition, the electrospray ionization mass spectrometric characterization has been employed to investigate the as-synthesized Au GSH clusters. However, because the spray is orthogonal to ion extraction in our utilized instrument, the clusters are unable to be detected in their native state and only certain fragments such as [Au(SG)$_2$-H]$^{-1}$ ($m/z$ 808) and [Au$_2$(SG)$_2$-H]$^{-1}$ ($m/z$ 1005) are detectable, as shown in Supplementary Fig. 2, which are in good agreement with the previous reports[34,35].

**Enhancing the photostability of Au GSH clusters**. The insulative SiO$_2$ spheres with rather low optical absorption

(Supplementary Fig. 3) have been chosen as inert supports to predominantly focus on investigating the photosensitizer role of Au GSH clusters in photocatalytic applications. The schematic synthesis procedure for SiO₂-Au GSH clusters composites via interfacial modification process has been displayed in Fig. 1a. The SiO₂ spheres become positively charged (Supplementary Fig. 4a) after surface modification with BPEI[36], a conjugated reducible

dendrimer, which leads to a substantial electrostatic attraction with negatively charged Au GSH clusters (Supplementary Fig. 4c) by Coulombic forces, thereby forming SiO₂-Au GSH clusters-BPEI composites (SAB) with strong interfacial interaction between Au GSH clusters and SiO₂ supports. In contrast, the common protocol used for loading Au GSH clusters onto the supports involves a pH value adjusted process under acidic

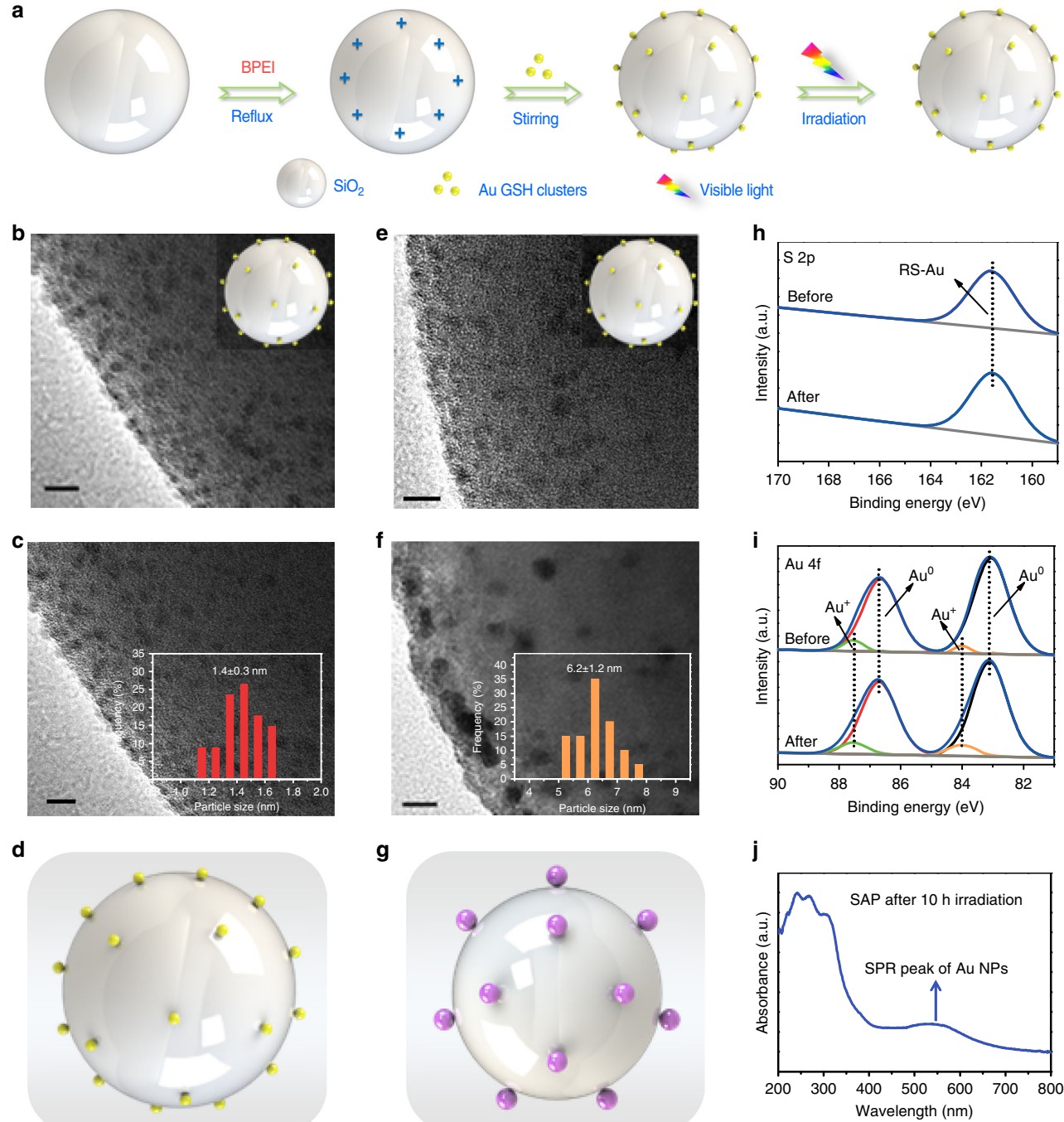

**Fig. 1** Photostability test and characterizations. **a** Schematic illustration of synthesis procedure for SiO₂-Au GSH clusters-BPEI composites (SAB) and photostability testing of as-prepared SAB; transmission electron microscopy (TEM) images of SAB **b** before and **c** after visible light irradiation (λ > 420 nm) for 10 h and SiO₂-Au GSH clusters-pH (SAP) **e** before and **f** after visible light irradiation for 10 h; the corresponding models of **d** SAB and **g** SAP after visible light irradiation for 10 h; high-resolution X-ray photoelectron spectroscopy (XPS) spectra of **h** S 2p and **i** Au 4f for SAB before/after visible light irradiation; **j** UV-vis diffuse reflectance spectrum (DRS) spectrum of SAP after 10 h visible light irradiation. Insets in **b** and **e** are the corresponding models of SAB and SAP before visible light irradiation. The histograms in **c** and **f** correspond to the particle size distributions of Au GSH clusters in the SAB and SAP after visible light irradiation for 10 h, respectively. The yellow spheres in **d** represent Au GSH clusters and the purple spheres in **g** represent Au nanoparticles. Scale bar, 5 nm

conditions[15,19,20,22,23,27]. As illustrated in Supplementary Fig. 5, SiO$_2$ dispersion is firstly adjusted to acidity (pH value at about 2) by adding dilute hydrochloric acid solution and then the Au GSH clusters solution is added into the above dispersion for fabricating SiO$_2$-Au GSH clusters-pH composites (SAP). Notably, due to the protonation of carboxylic groups in the glutathione ligands under acidic conditions[19] and the relatively weak electropositivity of SiO$_2$ spheres (Supplementary Fig. 4b), the as-prepared sample of SAP generally exhibits a weak Au–oxide interaction. The difference in the interaction between SiO$_2$ and Au GSH clusters over SAB and SAP samples may lead to the different loading amounts of Au GSH clusters on SiO$_2$ surface, which has been investigated by the inductively coupled plasma optical emission spectroscopy. As revealed in Supplementary Table 1, the addition amount of Au GSH clusters in SAB (0.86%) is higher than that of SAP with 0.28% Au GSH clusters, which could be attributed to the strong coulombic force between positively charged BPEI-SiO$_2$ spheres and the negatively charged Au GSH clusters. The XRD patterns of SAB and SAP samples in Supplementary Fig. 6 suggest the similar crystal structure of these composites and the diffraction peaks of Au GSH clusters are hardly observed for both of SiO$_2$-Au GSH clusters composites, which may be due to the low loading amount and poor crystallinity of Au GSH clusters[15,23,24].

To initially investigate the photostability of Au GSH clusters, the composites of SAB and SAP have been exposed to continuous visible light irradiation ($\lambda > 420$ nm) for 10 h under ambient conditions. TEM analysis suggests that Au GSH clusters maintain almost the same size on the surface of SiO$_2$ spheres for SAB composites before and after photo-irradiation (Fig. 1b–d), while ~1.4 nm Au GSH clusters aggregate to around 6 nm Au nanoparticles (NPs) for SAP samples after photo-irradiation (Fig. 1e–g), indicating that the aggregation of Au GSH clusters in the SAB samples has been effectively inhibited by BPEI interfacial modification, which is significantly different from that of SAP. The UV-vis diffuse reflectance spectrum (DRS) of SAP after light illumination in Fig. 1j shows a pronounced surface plasmon resonance (SPR) peak located at around 540 nm belonging to the metallic Au NPs[15,23,27], which consists with the TEM image of SAP sample after 10 h irradiation of visible light in Supplementary Fig. 7, further suggesting the seriously photoinduced aggregation of Au GSH clusters into Au NPs on the surface of SiO$_2$ spheres over SAP. Additionally, the TEM images in Supplementary Fig. 8 suggest that the size of Au GSH clusters over SAB after 24 h visible light irradiation ($\lambda > 420$ nm) is 1.4 nm, which is nearly unchanged as compared to that before light illumination (Fig. 1b), indicating that the interfacial modification of BPEI layer over SAB can provide a long-time protection (24 h) with regard to the stabilization of Au GSH clusters. When irradiation time over SAB is further prolonged to 36 and 48 h, the size of Au GSH clusters will increase to 2.0 and 2.1 nm, respectively, as illustrated in Supplementary Fig. 9, which indicates the slight aggregation of Au GSH clusters. Such increase in the size of Au GSH clusters may be attributed to the partial depletion of BPEI, since the BPEI layer that serves as a reducing agent would undergo oxidation and/or decomposition in the way to stabilize the Au GSH clusters. To further investigate the role of BPEI in inhibiting the growth of Au GSH clusters, the SAP sample is modified by BPEI (BPEI-SAP) and the obtained BPEI-SAP composite is irradiated under visible light ($\lambda > 420$ nm) for 10 h. The HRTEM image of BPEI-SAP after light illumination in Supplementary Fig. 10a suggests that, as compared with SAP samples (Fig. 1e–g), the serious aggregation of Au GSH clusters can be prevented to some extent and the size of Au GSH clusters is calculated to be 2.0 nm (Supplementary Fig. 10b).

To ensure whether the stability enhancement of Au GSH clusters due to the modification of BPEI is universal, we have performed a series of control experiments including the effects of various metal oxide supports and the structure of Au GSH clusters. The surfaces of different metal oxide supports, including rutile TiO$_2$, ZnO, and ZrO$_2$, have been positively charged by the BPEI modification (Supplementary Fig. 11), which can subsequently interact with the negatively charged Au GSH clusters (Supplementary Fig. 4c) via an electrostatic self-assembly method to produce metal oxide-Au GSH clusters-BPEI composites (MABs) and the photostability of MABs has been investigated under the same conditions as that for SAB and SAP. As illustrated in Supplementary Fig. 12, the mean diameter of Au GSH clusters on the surfaces of various metal oxide supports after continuous visible light ($\lambda > 420$ nm) for 10 h is determined to be 1.4 nm, indicating the critical role of BPEI on inhibiting the aggregation of Au GSH clusters and excluding the supports effect on the photostability enhancement of Au GSH clusters. Additionally, Supplementary Note 1 elucidates that, even though the Au GSH clusters suffer from serious aggregation on the surface of anatase TiO$_2$ due to the presence of abundance surface hydroxyl group, the presence of BPEI layer can protect ultrasmall Au GSH clusters from being fusion to some extent under visible light illumination, suggesting the universality of the interficial modification strategy on stabilizing Au GSH clusters. To further confirm our inference and achieve a generally valid conclusion, the Au$_{25}$-glutathione clusters (Au$_{25}$(SG)$_{18}$), which are well known as the most stable one among the thiolated clusters of this class[15,37,38], have been decorated on the surface of BPEI modified SiO$_2$ spheres to produce SiO$_2$-Au$_{25}$(SG)$_{18}$ clusters-BPEI composites (SASB). The photostability test in Supplementary Fig. 13 suggests that the size of Au$_{25}$(SG)$_{18}$ clusters can be well maintained, thereby demonstrating that the introduction of BPEI layer for interficial modification over SASB is an effective strategy to impede the photoinduced coalescence of Au$_{25}$(SG)$_{18}$ clusters to larger metallic Au NPs upon visible light illumination ($\lambda > 420$ nm).

To understand the role of BPEI in improving the stability of Au GSH clusters in SAB, X-ray photoelectron spectroscopy (XPS) analysis has been conducted. Survey XPS spectra in Supplementary Fig. 14d evidence the presence of Au, S, N, C, O and Si elements in the SAB composites before/after visible light irradiation. As mirrored in Fig. 1i, the two doublet 4f peaks in the high-resolution spectrum of Au 4f suggest that the there are two different elemental chemical states of Au species. The Au 4f$_{5/2}$ and Au 4f$_{7/2}$ peaks with binding energies of 86.9 and 83.1 eV are ascribed to metallic Au (Au$^0$), and the other doublet at 87.75 and 84.05 eV for Au 4f$_{5/2}$ and Au 4f$_{7/2}$ are assigned to Au$^+$ ion[39–41]. The Au$^0$ content determined by XPS is found to constitute ~90% of all Au atoms. The samples of SAB before/after photo-irradiation exhibit similar Au 4f spectra, demonstrating the well reserved structural properties of Au GSH clusters during the visible light irradiation ($\lambda > 420$ nm). Additionally, previous work has verified that the Au GSH clusters aggregation procedure is accompanied with a ligand oxidation process of Au GSH clusters, which results in a new S 2p peak matching the binding energy of R–SO$_3$ species (169.2 eV)[15]. As for the SAB composites after photo-irradiation of 10 h, only one S 2p$_{3/2}$ peak located at 161.5 eV (Fig. 1h), corresponding to the ligand features (RS–Au) in various monothiolate Au GSH clusters[42,43], is observed, which is similar with that of SAB sample before light irradiation, demonstrating that the organic ligand oxidation process over the Au GSH clusters in SAB has been efficiently inhibited during visible light irradiation. The enhanced photostability of Au GSH clusters in the SAB composites can be attributed to the introduction of BPEI with high reductive ability[44–46], which could serve as effective reductive agents to hamper the ligand oxidation process over Au GSH clusters during photo-irradiation process. As illustrated in Supplementary Fig. 14a–c, the slight

peak shift of the elements of C, O, and N over the SAB sample before/after the visible light irradiation clearly suggests the redox process of BPEI to protect organic ligands from being oxidized and thus enhancing the photostability of Au GSH clusters in the SAB composites. Aiming to in depth explore the underlying mechanism of enhanced photostability of Au GSH clusters over SAB composites, the surface charge modifying agent of 3-aminopropyltriethoxysilane (APTES), which can also positively charge the SiO₂ spheres surface[47], but with no reductive ability, has been employed to replace BPEI for interfacial modification. After coupling with negatively charged Au GSH clusters, the as-formed SiO₂-Au GSH clusters-APTES composites (SAA) have been exposed under visible light irradiation ($\lambda > 420$ nm) for 10 h. TEM images of SAA after light irradiation (Supplementary Fig. 15a, b) suggest that the Au GSH clusters (1.4 nm) aggregate into larger Au NPs (3.1 nm) (Supplementary Fig. 15c), indicating that the APTES as a surface charge modifying agent cannot protect the Au GSH clusters from being fusion. The structural formulas of APTES and BPEI in Supplementary Fig. 16 suggest that different from APTES, BPEI contains abundant primary, secondary and tertiary amino groups. It has been well documented that the multiple polymer chains of primary, secondary and tertiary amine groups in the BPEI can encapsulate the as-synthesized Au GSH clusters via crosslinking[45], which could fix these clusters on the surface of supports and hamper the migration of clusters, thus stabilizing the ultrasmall Au GSH clusters. Accordingly, based on the above control experiments, we can deduce that it is the high reductive ability and branched structure of BPEI that play a critical role in hampering the ligand oxidation process over Au GSH clusters, thereby stabilizing Au GSH clusters against aggregation under continuous light illumination condition.

The photocatalytic performances of bare SiO₂ spheres and SiO₂-Au GSH clusters-BPEI (SAB) composites have been tested by degradation of organic dye rhodamine B (RhB), a typical nonselective photocatalytic processes[48–50], under visible light irradiation ($\lambda > 420$ nm). As shown in Supplementary Fig. 17, SiO₂ spheres exhibit negligible photoactivity toward RhB degradation during 10 h visible light irradiation, which suggests that SiO₂ spheres are not photocatalytically active under visible light irradiation and the dye photosensitization effect of RhB[48] on the photoactivity can be ignored. After coupling with Au GSH clusters as visible light photosensitizers, the SAB sample shows obvious activity toward photocatalytic RhB degradation (Supplementary Fig. 17). The observed photoactivity can be attributed to the incorporation of Au GSH clusters, which can generate electron–hole pairs by electron transitions from the highest occupied molecular orbital (HOMO) to the lowest unoccupied molecular orbital (LUMO) under visible light irradiation[23,51,52], thereby driving RhB degradation. However, it should be noted that only 38% of RhB has been degraded after 10 h irradiation of visible light over SAB, indicating the low photosensitization efficiency of Au GSH clusters. Such observed low photoactivity could be attributed to large band gap of SiO₂ support[47], by which the photogenerated electrons in the LUMO of Au GSH clusters cannot transfer to the conduction band (CB) of SiO₂ efficiently, thereby leading to simultaneously serious recombination of charge carriers. Notably, it has been demonstrated that the CB edge potential of metal oxides (e.g., ZnO and rutile TiO₂) is more positive than the LUMO potential of Au GSH clusters, which enables the transformation of photoexcited electrons from Au GSH clusters to the metal oxide supports[15,24,53]. The photoactivities of BPEI modified metal oxides-Au GSH clusters composites (denoted as MAB) have been evaluated toward RhB degradation under visible light irradiation ($\lambda > 420$ nm). As shown in Supplementary Fig. 18, the samples of MAB exhibit

moderate photoactivity enhancement than bare semiconductors, which suggests that the random loading of ultrasmall Au GSH clusters onto semiconductors without rational structure design is inefficient to achieve high efficient Au GSH clusters-semiconductor composites. Therefore, a thickness tunable TiO₂ shell has been coated onto the surface of SAB composites for designing core-shell SiO₂-Au GSH clusters-BPEI@TiO₂ structures to construct high performance Au GSH clusters-semiconductor composites for solar energy conversion.

**Elaboration of SiO₂-Au GSH clusters-BPEI@TiO₂ samples.** The synthesis procedure of core-shell SiO₂-Au GSH clusters-BPEI@-TiO₂ (SABT) composites has been illustrated in Fig. 2a. In brief, the SAB composites are firstly dispersed in ethanol, followed by adding hexadecylamine (HDA) surfactants and ammonia under vigorously stirring at room temperature. The HDA surfactants could segregate the surface of SAB. Then, different amounts of titanium isopropoxide (TIP) have been introduced into the dispersion under stirring. During this process, TIP hydrolysis products (i.e., TiO₂) participate in hydrogen-bonding interactions with the amino groups of HDA molecules to form inorganic–organic composites that coat the SAB spheres to produce SABT[54]. Scanning electron microscopy (SEM) images of SABT composites in Supplementary Fig. 19 show the uniform spherical morphology of these SABT composites. TEM analysis in Fig. 2 shows that the core-shell structures of SABT composites are well defined, and the thickness of TiO₂ shell can be well-tuned from few to dozens nanometer by adjusting the addition amounts of TIP from 0.05 to 0.2 mL (marked as SABT-$x$, where the $x$ represents the volume of added TIP). In particular, the Au GSH clusters embedded into TiO₂ shell are distinguishable, especially for the samples of SABT-0.05 (Fig. 2c) and SABT-0.1 (Fig. 2d) with thin TiO₂ shell. Noteworthily, the Au GSH clusters in core-shell SABT composites maintain their original sizes, suggesting the stability of Au GSH clusters during such a mild TiO₂ coating process at room temperature. The energy-dispersive X-ray (EDX) spectrum in Fig. 2h evidences the presence of Au, O, Si, and Ti elements over SABT-0.05 composites. Additionally, the elemental mapping analysis identifies the spatial distribution of these elements in SABT-0.05. As displayed in Fig. 2b, e, the distribution ranges of Ti and O elements are slightly larger than that of Au, which is itself slightly larger than that of Si. These corroborate the intelligently designed core-shell SABT structure, among which the thin TiO₂ layer is located at the outmost surface, while the SiO₂ spheres serve as the core and the Au GSH clusters are located between the SiO₂ core and TiO₂ shell.

The phase composition and optical properties of the SABT composites have been characterized by XRD and DRS analysis, respectively. As reflected in Supplementary Fig. 20, all of the samples possess analogous XRD patterns with a broad peak located at ca. 23°, corresponding to amorphous silica[55], and the intensities of SiO₂ diffraction peaks become weak with increasing the thickness of TiO₂ shells. However, no typical diffraction peaks of TiO₂ have been observed in XRD spectra because TiO₂ is amorphous by TIP hydrolysis[54]. Due to the relatively low content and weak diffraction intensity of Au GSH clusters, the diffraction peak of Au GSH clusters in the SABT has also not been perceived[15,23–28]. The DRS spectra of SABT composites in Supplementary Fig. 21 show that the introduction of TiO₂ shell can greatly increase the UV absorption intensities of the samples, while the visible light absorptions in the range of 400–520 nm caused by Au GSH clusters almost maintain their intensities.

**Photocatalytic activity and stability of SABT composites.** The photocatalytic performances of SABT composites have been

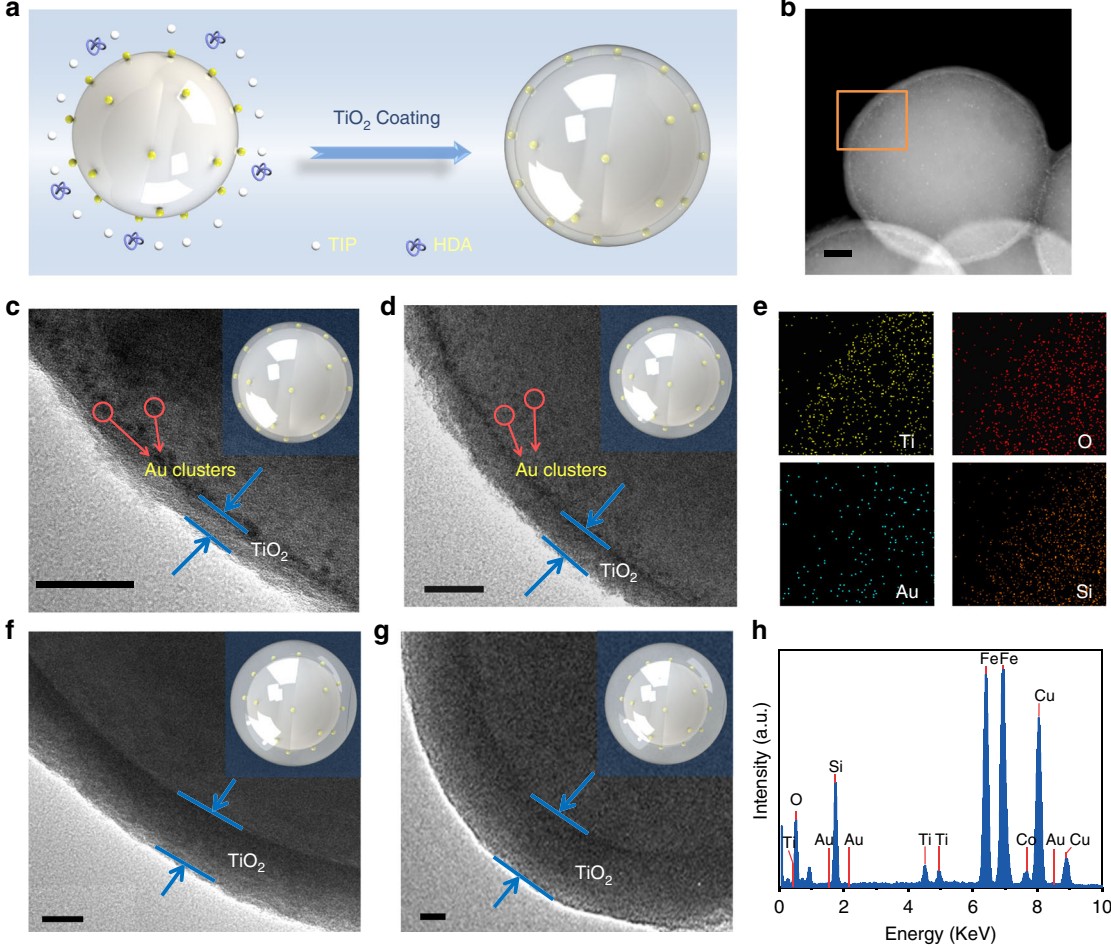

**Fig. 2** Morphology and chemical composition of SABT samples. **a** Schematic illustration of fabrication process for core-shell SiO$_2$-Au GSH clusters-BPEI@TiO$_2$ (SABT) composites; **b** high-angle annular dark-field scanning transmission electron microscopy (HAADF-STEM) image (scale bar, 50 nm); **e** elemental mapping results; and **h** energy-dispersive X-ray (EDX) spectrum for the boxed area in the main image of SABT-0.05; TEM images of **c** SABT-0.05, **d** SABT-0.1, **f** SABT-0.15, and **g** SABT-0.2, scale bar, 20 nm. Insets in the **c**, **d**, **f**, and **g** are the corresponding models of SABT with different TiO$_2$ shell thickness

evaluated toward rhodamine B (RhB) photodegradation under visible light illumination ($\lambda > 420$ nm). As delineated in Fig. 3a, the photoactivity over SAB sample can be significantly enhanced by coating TiO$_2$ shell and the dependence of RhB degradation efficiency on the thickness of the TiO$_2$ shell shows a clear volcano behavior. The photocatalytic degradation of RhB firstly increases with increasing thickness of TiO$_2$ layer, and the maximum RhB degradation efficiency has been achieved over SABT-0.15 and then decreases. Specifically, 99.1% conversion of RhB is achieved over SABT-0.15 after visible light irradiation for 0.5 h, and the photocatalytic efficiency of SABT-0.15 is about 19 times higher than that of SAB with only 4.9% conversion of RhB under identical reaction conditions. Furthermore, the sample of TiO$_2$-Au GSH cluster-BPEI (TAB) exhibits worse catalytic activity than the SABT composites, which indicates that the intelligently designed core-shell structure of SABT is favorable for enhancing the photocatalytic performance of Au GSH clusters-semiconductor composites. The pseudo-first order kinetic of the degradation of RhB based on the above data is shown in Supplementary Table 2. It is clear that all of the SABT samples show higher reaction rate than the SAB and TAB samples, among which SABT-0.15 exhibits the highest reaction rate of 0.119 min$^{-1}$ among these SABT composites. Additionally, the photocatalytic degradation of RhB over SiO$_2$-Au GSH clusters-pH@TiO$_2$ composites (SAPT) with different TiO$_2$ shell thickness has also been

investigated. Due to the addition amount of Au GSH clusters in SAP is different from that of SAB (Supplementary Table 1), the photocatalytic activities of RhB degradation over various SAPT composites with different TiO$_2$ shell thickness have been normalized with respect to the loading amount of Au GSH clusters in SAB for a fair comparison. As displayed in Supplementary Fig. 22, the coating of TiO$_2$ shell greatly improves the photocatalytic efficiency of SAP sample and the SAPT-0.15 composites exhibit the best photoactivity among these samples. However, the RhB degradation efficiency over SAPT-0.15 composites is worse than that of SABT-0.15 (Fig. 3a) under identical reaction conditions, which suggests that the presence of BPEI layer is beneficial for the photoactivity enhancement since the photostability of Au GSH clusters in the SABT-0.15 is significantly ameliorated.

To investigate the effect of surface coating of TiO$_2$ shell on the photocatalytic performance and exclude the possible dye photo-sensitization effect of RhB on the activity[56], the visible light photoactivities of SiO$_2$ spheres, TAB, SAB, and SABT composites toward photocatalytic reduction of $p$-methoxy nitrobenzene to $p$-methoxy aniline have been evaluated, which is a typical six-electron reduction reaction (Supplementary Fig. 23)[24,27,47,57]. As presented in Fig. 3b, the higher photocatalytic activity of SABT composites than that of SiO$_2$ spheres and SAB sample verifies the positive role of TiO$_2$ shell in improving the photoactivity of SiO$_2$-

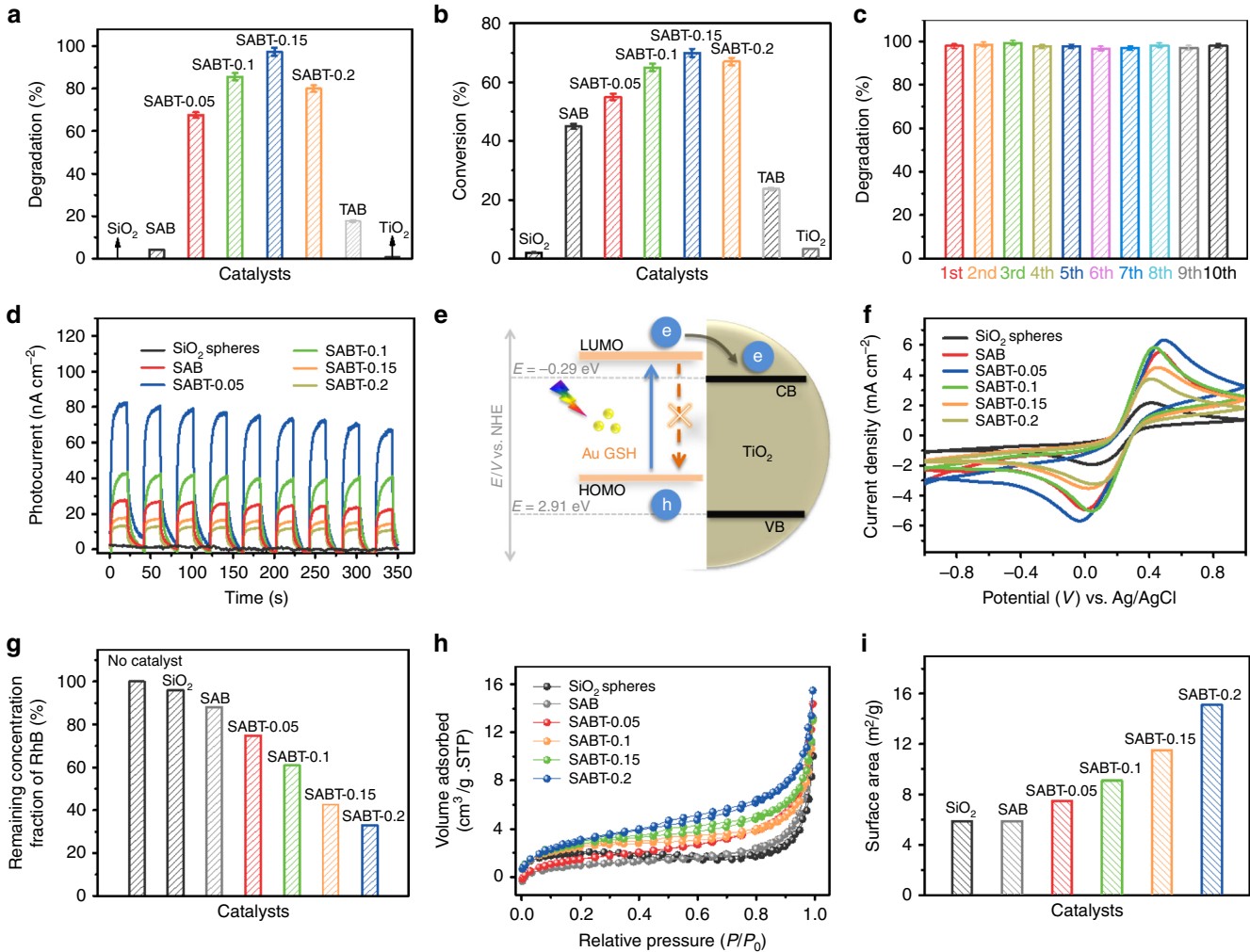

**Fig. 3** Photocatalytic activity and the underlying mechanism Photocatalytic degradation of **a** RhB over blank SiO₂ spheres, TiO₂, SAB, TAB, and SABT composites with different TiO₂ shell thickness under visible light irradiation (λ > 420 nm) for 0.5 h; photocatalytic reduction of **b** *p*-methoxy nitrobenzene to *p*-methoxy aniline over blank SiO₂ spheres, TiO₂, SAB, TAB, and SABT composites with different TiO₂ shell thickness under visible light irradiation for 5 h; **c** recycling photocatalytic degradation of RhB over optimal SABT-0.15 composite under visible light irradiation (λ > 420 nm); **d** transient photocurrent densities of SiO₂ spheres, SAB and SABT composites with different TiO₂ shell thickness under visible light irradiation (λ > 420 nm); **e** scheme illustrating the photogenerated electron transport pathways between Au GSH clusters and TiO₂ shell (the blue arrow represents the photoexcitation process of electron–hole pairs; the orange dash line means the recombination of electron-hole pairs; e and h in the blue cycles correspond to the photogenerated electron and hole, respectively); **f** cyclic voltammograms of SiO₂ spheres, SAB and SABT composites with different TiO₂ shell thickness; **g** bar plots showing the remaining RhB in reaction solutions after being kept in dark for 3 h to achieve the adsorption–desorption equilibrium over SiO₂ spheres, SAB and SABT composites with different TiO₂ shell thickness; **h** nitrogen adsorption–desorption isotherms of SiO₂ spheres, SAB and SABT composites with different TiO₂ shell thickness; **i** surface area of SiO₂ spheres, SAB and SABT composites with different TiO₂ shell thickness. Note that the error bars represent the photoactivity s.d. values calculated from triplicate experiments

Au GSH clusters composites. And the SABT-0.15 shows the best photoactivity among these samples, elucidating that the thickness of TiO₂ shell plays a critical role in determining the photocatalytic performance of SABT toward *p*-methoxy nitrobenzene reduction. Moreover, due to the intelligently designed core-shell structure, all of the SABT composites show higher catalytic activity than the TAB composite.

In addition to the TiO₂ shell thickness, the crystallization of TiO₂ layer in composites is also a factor that affects the photocatalytic performance of SABT composites and it has been reported that the amorphous TiO₂ usually exhibits poor photocatalytic activity[58–62]. We have paid our endeavors to crystallize TiO₂ shell in SABT-0.15, including calcination and hydrothermal treatment, since the high temperature is generally required to transform amorphous TiO₂ particle into an anatase

one. Even though the crystallization of TiO₂ can be improved by calcination and hydrothermal treatment, the Au GSH clusters in SABT-0.15 suffer from serious instability under high temperature and aggregate into metallic Au nanoparticles with large size, as displayed in Supplementary Fig. 24, 25, thereby deteriorating the photocatalytic performance of SABT-0.15 sample toward various photocatalytic reactions under visible light illumination (λ > 420 nm) (Supplementary Fig. 26).

The photostability of the optimal SABT-0.15 has been evaluated via recycling experiments, as shown in Fig. 3c. The apparent activity loss has not been observed for RhB photodegradation during ten successive recycle runs, indicating that SABT-0.15 is able to serve as a stable visible light photocatalyst. Unfortunately, the size information of Au GSH clusters in SABT-0.15 after visible light illumination cannot be directly obtained

due to the thick TiO₂ shell. Alternatively, the SABT-0.05 composite with a thin layer of TiO₂ has been selected as a representative sample for observing the size of Au GSH clusters after photocatalytic reactions. The TEM image (Supplementary Fig. 27) and EDX spectrum (Supplementary Fig. 28) of SABT-0.05 irradiated in RhB solution under visible light for 4 h suggest that the Au GSH clusters maintain their particle size, verifying the excellent photostability of Au GSH clusters in SABT-0.05.

**Mechanism of the photoactivity improvement**. To understand the origin of photoactivity enhancement of SABT compared to SAB sample and the effects of TiO₂ shell thickness on photo-activities of SABT composites, we have investigated the structure-photoactivity relationship in terms of joint analysis on the charge separation efficiency and surface area. The photoelectrochemical (PEC) analysis has been performed to investigate the charge carrier transfer process over SiO₂ spheres, SAB and SABT with different TiO₂ shell thickness. Figure 3d shows the periodic on/off transient photocurrent response of these samples under the intermittent visible light photo-irradiation. Bare SiO₂ spheres show no photocurrent response under visible light irradiation ($\lambda$ > 420 nm), which is consistent with its large band gap[47]. After coupling with Au GSH clusters, the obvious photocurrent density over SAB verifies the substantial photosensitization effect of Au GSH clusters on the PEC performance of SAB in the range of visible light[15,63,64]. With the coating of TiO₂ thin layer, the SABT-0.05 sample exhibits higher photocurrent density than SAB, which suggests that the separation of photoinduced charge carriers over SABT-0.05 has been efficiently facilitated as compared to that of SAB. However, a thicker TiO₂ shell leads to the decreased photocurrent densities of SABT composites, which may be attributed to the fact that the thicker layer of TiO₂ will block the transport of photogenerated electrons from Au GSH clusters to the back contact, thus reducing the overall photocurrent response. The cyclic voltammograms (CV) curves of bare SiO₂ spheres, SAB and SABT composites in Fig. 3f reveal that the introduction of thin TiO₂ layer can enhance the current densities of SABT composites as compared to that of SAB, while the thick TiO₂ layer results in decreased current densities, which is consistent with the transient photocurrent results in Fig. 3d. Electrochemical impedance spectroscopy (EIS), as a method to monitor charge transfer process on the electrode and at the contact interface between electrode and electrolyte[64], has further confirmed that the SABT with thin shell of TiO₂ layer improves the charge carrier migration of SABT composites efficiently as compared to that with thick one (Supplementary Fig. 29). The above PEC experiments together confirm that the coating of thin TiO₂ layer is beneficial for efficiently separating and transferring the photogenerated electrons from Au GSH clusters (Fig. 3e), hampering the recombination of charge carriers, and thereby resulting in the higher photocatalytic efficiency of SABT composites than that of SiO₂ spheres and SAB under visible light illumination ($\lambda$ > 420 nm).

The nitrogen (N₂) adsorption–desorption isotherms have been performed to investigate the surface area of blank SiO₂ spheres, SAB and SABT composites. As illustrated in Fig. 3h, i and Supplementary Table 3, the loading of Au GSH clusters leads to little variation of surface area of bare SiO₂ spheres, while the surface area of the SABT is gradually improved by increasing the thickness of TiO₂ shell. Adsorption experiments toward RhB (Fig. 3g) and *p*-methoxy nitrobenzene (Supplementary Fig. 30) in the dark suggest that the SABT composites exhibit improved adsorption capacity as compared to SiO₂ spheres and SAB sample. In addition, with the increased thickness of TiO₂ shell, the adsorption capacity of SABT composites increases, which is

consistent with the surface area of SABT composites (Fig. 3i). The above characterizations suggest that the primary role of coating TiO₂ semiconductor layer onto SAB for ameliorating the photocatalytic efficiency of SABT composites is two-fold. One is to improve the separation and migration efficiency of photo-induced charge carriers from excited Au GSH clusters. The other is to enhance the adsorption capacity of the catalysts toward the reactants. By adjusting the thickness of TiO₂ shells, the photoactivity of SABT can be tuned due to cooperative adsorption and PEC properties. Therefore, the optimal SABT-0.15 composites exhibit the highest activity toward photocatalytic degradation of RhB among these SABT samples.

Control experiments using different radical scavengers help us to further understand the underlying reaction mechanism of the degradation of RhB over SABT-0.15. As shown in Supplementary Fig. 31, the addition of benzoquinone (BQ) scavenger for superoxide radicals[65] almost terminates the photocatalytic reaction of RhB degradation. A similar and obvious inhibition phenomenon for photocatalytic reaction is also observed, when *tert*-butyl alcohol (TBA) scavenger for hydroxyl radicals[66], ammonium formate (AF) scavenger for holes[67], and potassium persulfate (K₂S₂O₈) scavenger for electrons[68] are added, indicating that the hydroxyl radicals, holes, electrons, and superoxide radicals all show a significant impact on the degradation of RhB. Nevertheless, the inhibition degree of photoactivity induced by the addition of TBA, AF, and K₂S₂O₈ is smaller than the case of BQ, suggesting that the superoxide radicals play a more important role than other reactive species toward photodegradation of RhB over SABT-0.15. Taking K₂S₂O₈ as scavenger for photogenerated electrons[68], the control experiments for photo-reduction of *p*-methoxy nitrobenzene under visible light illumination ($\lambda$ > 420 nm) have been performed (Supplementary Fig. 32). The addition of K₂S₂O₈ greatly inhibits the reduction of *p*-methoxy nitrobenzene, which confirms that the reduction reactions are driven by the photoexcited electrons. Supplementary Fig. 33 shows the electron spin resonance (ESR) spectra of superoxide radicals detected over the samples, which suggests that the coating of thin TiO₂ shells contributes to improving the amounts of such reactive oxidative species (ROSs) generated over SAB. This is consistent with the more efficient separation and transfer of charge carriers over SABT than that over SiO₂ spheres and SAB. Although the onslaught of ROSs on organic ligands of Au GSH clusters can lead to the instability of Au GSH clusters and their aggregation into large Au nanoparticles[15], the following factors are considered to be the decisive role for achieving high photostability of Au GSH clusters in SABT composites. On one hand, as discussed above, BPEI has high reductive capability[44,45], which can primitively protect the organic ligands of Au GSH clusters from being oxidized during the photo-irradiation process. On the other hand, the intelligent design of core-shell SABT composites by coating TiO₂ shell can enhance the adsorption capacity of catalysts toward reactants, which would consume the ROSs timely and effectively reduce the possibility of the reactions between ROSs with the organic ligands of Au GSH clusters, thus leading to the improvement of the photostability of Au GSH clusters in SABT composites.

**Discussion**
Although prior literatures have already elucidated the coalescence and agglomeration mechanism of ultrasmall Au GSH clusters, it still remains a challenging task to achieve highly stable Au GSH clusters on the substrates under in situ photo-irradiation conditions, which significantly impedes the practical implementations of Au clusters-based catalysts systems. In this work, the utilization of multifunctional BPEI as a surface charge modifying, reducing

and stabilizing agent for interfacial modification of $SiO_2$-Au GSH clusters composites has demonstrated to be an efficient and universal strategy for protecting the Au GSH clusters from being fusion during longtime light illumination. Such photostability enhancement is ascribed to the high reductive ability of BPEI, by which the organic ligand oxidation process over Au GSH clusters can be inhibited, and this in turn results in the improvement of durability. Additionally, while simultaneously maintaining the photostability of Au GSH clusters, the sequential coating of thickness tunable $TiO_2$ shells for constructing core-shell $SiO_2$-Au GSH clusters-BPEI@$TiO_2$ (SABT) nanostructures has been demonstrated to significantly ameliorate the photoactivity by regulating the photoelectrochemical and adsorption properties of SABT composites synergistically.

In summary, we have reported a combined strategy via interfacial modification and composition engineering for enhancing the long-term stability of ultrasmall Au GSH clusters with improved photocatalytic performance, which is expected to provide a facile guideline for rationally structural design of Au clusters-based composites toward various targeting applications in solar energy conversion.

## Methods

**Materials**. Gold(III) chloride trihydrate ($HAuCl_4\cdot3H_2O$), acetonitrile ($C_2H_3N$, HPLC grade), sodium hydroxide (NaOH), hydrochloric acid (HCl), tetraethyl orthosilicate ($C_8H_{20}O_4Si$, TEOS), isopropanol ($C_3H_8O$), ethanol ($C_2H_6O$), ammonia ($NH_3\cdot H_2O$), p-methoxy nitrobenzene ($C_7H_7NO_3$), rhodamine B ($C_{28}H_{31}ClN_2O_3$, RhB); ammonium formate ($CH_5NO_2$), potassium persulfate ($K_2S_2O_8$), tert-butyl alcohol ($C_4H_{10}O$), p-benzoquinone ($C_6H_4O_2$) were supplied by Sinopharm Chemical Reagent Co., Ltd. (Shanghai, China). L-glutathione reduced ($C_{10}H_{17}N_3O_6S$, GSH), hexadecylamine ($C_{16}H_{35}N$, HDA), branched polyethylenimine ($M_w$ ~25,000, BPEI) and titanium isopropoxide ($C_{12}H_{28}O_4Ti$, TIP), 5,5-dimethyl-1-pyrroline-N-oxide ($C_6H_{11}NO$, DMPO) were obtained from Sigma-Aldrich. All materials were analytical grade and used as received without further purification. Deionized (DI) water used in the synthesis was from local sources.

**Catalyst preparation**. Synthesis of Au GSH clusters: Au GSH clusters were fabricated by procedures reported previously with modification[31]. Typically, 0.24 g of gold (III) chloride trihydrate were first dissolved in 200 mL DI water. Then, 100 mL DI water contained 0.276 g of L-glutathione were added into above solution under stirring. A colorless solution was obtained after 2 h stirring and the mixture was heated at 343 K in an oil bath for 24 h. After removing the heat source, the Au GSH clusters were recrystallized in acetonitrile and then thoroughly washed with DI water and acetonitrile (1: 3 in volume) for three times by centrifuging at 10,500×g for 5 min. A desired concentration of Au GSH clusters solution was obtained by redisperse the purified Au GSH clusters in DI water with assistance of few drops of 0.5 M NaOH and stored at 277 K for further use.

Synthesis of $SiO_2$ spheres: In a typical reaction, a mixture contained 13 mL of ammonia (25–28%), 63.3 mL of isopropanol and 23.5 mL of DI water was heated to 308 K in an oil bath. A volume of 0.6 mL of TEOS (99%) was introduced into this solution dropwise and the mixture was kept at 308 K for 30 min. Then, 5 mL of TEOS were added dropwise into the reaction system. After 2 h reaction, the $SiO_2$ spheres were separated by centrifugation, washed with ethanol and DI water repeatedly, and finally dried in air.

Synthesis of BPEI-modified $SiO_2$ spheres: A volume of 4 mL BPEI solution (86 mg mL$^{-1}$) was added into 200 mL ethanol with $SiO_2$ spheres concentration of 2 mg mL$^{-1}$. The mixtures were refluxed at 333 K in an oil bath for 4 h. The BPEI-modified $SiO_2$ spheres were washed with ethanol and DI water repeatedly, and finally dried in air.

Synthesis of $SiO_2$-Au GSH clusters-BPEI composite (SAB): 0.1 g positively charged BPEI-modified $SiO_2$ spheres were dispersed in 50 mL DI water by ultrasonication. Then, 5 mL negative charged Au GSH clusters (0.2 mg mL$^{-1}$) were added dropwise into this dispersion. After stirring for 1 h, SAB were isolated by centrifugation, washed with ethanol and DI water repeatedly, and finally dried in air.

Synthesis of core-shell $SiO_2$-Au GSH clusters-BPEI@$TiO_2$ nanostructures: 0.08 g of as-prepared SAB composites and 0.08 g of HDA (90%) were ultrasonically dispersed in 9.74 mL of ethanol. Then, 0.2 mL of ammonia was introduced into the solution under stirring. After 1 min, different amounts of TIP (97%) were added into the dispersion. After reaction for 10 min, the core-shell $SiO_2$-Au GSH clusters-BPEI@$TiO_2$ composites were collected by centrifugation and washed several times with DI water and ethanol[54]. The details for synthesis of $Au_{25}(SG)_{18}$ clusters, $SiO_2$-$Au_{25}(SG)_{18}$ clusters-BPEI composite (SASB), BPEI modified-metal oxide nanoparticles (ZnO and $ZrO_2$), metal oxide nanoparticles (ZnO and $ZrO_2$)-Au GSH clusters-BPEI composites (MABs) and $SiO_2$-Au GSH clusters-pH@$TiO_2$ nanostructures by hydrolysis of TIP are presented in Supplementary Methods.

**Characterization**. Zeta sizer 3000HSA was employed to determine the Zeta potentials ($\xi$) values of the samples at room temperature. The optical properties of the samples were measured by UV-vis spectrophotometer (Perkin Elmer Lambda 950 UV-Vis-NIR). The XRD patterns of different samples were collected by a Bruker D8 Advance X-ray diffractometer (40 kV, 40 mA) using Ni-filtered Cu Kα radiation. The SEM images were taken on FEI Nova NANOSEM 230 spectro-photometer. TEM, EDX, and elemental mapping results were obtained by a JEOL model JEM 2010 EX instrument. PL spectra of the samples were recorded on an Edinburgh FL/FS920 spectrophotometer. XPS measurement was performed on a Thermo Scientific ESCA Lab 250 spectrometer with monochromatic Al Kα as the X-ray source. The C1s peak at 284.6 eV was used to calibrate binding energies. The loading amounts of Au GSH clusters in the SAB and SAP samples were quantified by an inductively coupled plasma optical emission spectroscopy instrument (ICP-OES, PerkinElmer Optima 2000DV). The surface areas of various samples were obtained by the Micromeritics ASAP 3020 equipment. Mass spectra were recorded on an Agilent Technologies ESI-TOF-MS. ESR signal was recorded using a Bruker EPR A300 spectrometer.

Photoelectrochemical analysis was carried out in a three-electrode quartz cell, among which Ag/AgCl electrode and Pt plate were used as reference electrode and counter electrode, respectively. FTO glass was used as support to prepare working electrode. Specifically, the slurry prepared by dispersing 5 mg sample in 0.5 mL of DMF was spread onto FTO glass with exposed area of 0.25 cm$^2$. After air drying, the working electrodes were dried at 393 K for 2 h to enhance adhesion. BAS Epsilon workstation was used to measure the photocurrent density of samples in the electrolyte of 0.2 M $Na_2SO_4$ aqueous solution (pH = 6.8). CV and EIS experiments were carried out on Autolab, PGSTATM204 in the electrolyte of 0.5 M KCl aqueous solution containing 0.01 M $K_3[Fe(CN)_6]$/$K_4[Fe(CN)_6]$ (1:1) under open circuit potential conditions.

**Photoactivity testing**. Degradation of organic dye: Ten milligrams of catalyst were suspended in 40 mL of 10 mg L$^{-1}$ RhB solution. The mixture was stirred in dark for 3 h and then irradiated with a 300 W Xe arc lamp ($\lambda > 420$ nm) under ambient conditions. After photocatalytic reaction, 4 mL of sample solution were taken from reaction system and analyzed on a Varian UV-vis spectrophotometer (Cary 50, Varian Co.).

Reduction of aromatic nitro compound: Fifty milligrams of photocatalyst and 100 mg ammonium formate were dispersed in 30 mL of 5 mg L$^{-1}$ p-methoxy nitrobenzene solution. After stirring in dark for 3 h, the mixture was irradiated by visible light ($\lambda > 420$ nm) using a 300 W Xe arc lamp. After photocatalytic reaction, 4 mL of sample solution were taken from reaction system and analyzed on a Varian UV–vis spectrophotometer (Cary 50, Varian Co.).

**Data availability**. The data that support the findings of this study are available from the figshare repository with DOI number of 10.6084/m9.figshare.5938669.

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

## Acknowledgements

Y.-J.X. acknowledges the support from the National Natural Science Foundation of China (U1463204, 21173045), the Award Program for Minjiang Scholar Professorship, the NSF of Fujian Province for Distinguished Young Investigator Rolling Grant (2017J07002), the Independent Research Project of State Key Laboratory of Photocatalysis on Energy and Environment (No. 2014A05), the 1st Program of Fujian Province for Top Creative Young Talents, and the Program for Returned High-Level Overseas Chinese Scholars of Fujian province. H.M.C. appreciates the support from the Ministry of Science and Technology, Taiwan (Contracts No. MOST 106-2119-M-002-031).

## Author contributions

Y.-J.X. proposed the research direction and supervised the project. B.W. designed and preformed the experiments. Z.C.T. performed MS measurement and H.M.C. provided helpful suggestions in conducting the study. K.Q.L. carried out TEM characterization and analysis. Y.-J.X. and B.W. wrote and revised the manuscript. All authors participated in discussion and reviewed the manuscript before submission.

## Additional information

**Competing interests:** The authors declare no competing interests.

