## [Peer Review File · Nature Communications]

Reviewers' comments:

Reviewer #1 (Remarks to the Author):

Suppressing the aggregation of ultrasmall Au clusters during light irradiation is one of the important issues to be addressed in the application of Au clusters. This article reports a way of doing it by using poly-ethylenimine. At a quick glance, the role of this surface modification works very well and I think this strategy works to some extent. However, the enhanced performance in photocatalysis is not necessarily from the result of better stability. The apparent difference in the interaction between SiO₂ and Au clusters must result in different loading of Au clusters on SiO₂. Therefore, a quantitative analysis for the loading amount should be performed and the photocatalytic activity must be normalized with respect to the determined amount of Au clusters. Otherwise, a fair comparison cannot be made. And I am curious about the photostability of poly-ethylenimine. As stated in the manuscript, the authors claimed that BPEI can serve as a reducing agent to prevent the oxidation of the ligand of Au clusters, which implies that BPEI should be oxidized. This oxidative degradation is also anticipated given the decomposition of other organic substances demonstrated in this work. Hence, the protection capability of BPEI may be restricted if the irradiation is further prolonged. This possibility needs to be investigated. SABT-0.15 is claimed as the best catalyst among other counterparts for the decomposition of RhB and the reduction of Cr(IV) and p-methoxy nitrobenzene. However, the results from the PEC measurements don't comply with this trend. SABT-0.05 works best, which is attributed to the thin TiO₂ layer. With the thicker TiO₂ coating, the catalytic activity get lower due to the increased recombination. These observations are somewhat confusing. Both experimental situations are different, but in terms of charge recombination they share the same recombination dynamics. This work looks systematically performed and the analyses are quite thorough. The idea of using BPEI is interesting, but I do not think the significance of this work meets the requirement for the publication in this journal. Scientific Reports or other relevant journals would be a better place for this work if all the concerns are properly addressed.

Reviewer #2 (Remarks to the Author):

The manuscript on preparation of highly stable gold nanoclusters by B. Weng et al. is very interesting. Although, the overall quality of the manuscript is high, there are some important issues that should be explained before possible publications, as shown below:

1) The mechanism must be clarified. Authors proposed photo-generation of electrons (able to

form reactive oxygen species) and high stability of the system. This two contradict propositions must be explained, i.e., gold should be oxidized by this mechanism, and thus either it can be dissolved or electron donor should be applied (considering glutathione as electron donor, its regeneration should be proposed),

2) Dyes cannot be used for activity testing due to sensitization mechanisms (see reviews by B. Ohtani), other colorless compounds should be used for mechanism study (Application of chromate as testing molecule is also not recommended since it could sensitize titania (as has been published, e.g., Kuncewicz et al.)).

3) Authors concluded that titania played important role in the mechanism (this is obvious), but they did not perform any reference experiments for: (i) bare titania and (ii) titania with gold nanoclusters.

4) What about photocatalytic activity of ZnO-Au GSH clusters-BPEI (shown in Supplementary Figure 9)?

5) Although, the existence of optimal thickness of titania is not surprising, especially in the case of amorphous titania with plenty of recombination sites (and many reports on similar behaviors have been already published; some of them should be cited), there is a question why crystallization of titania was not performed.

6) Authors presented plenty of supporting data, but some of them were not discussed, e.g., interesting data shown in Supplementary Figure 22 (impedance spectroscopy) should be discussed (at least in supplementary material file) and values on Y-axis should be shown.

7) Do authors use “Zeta potential” and “surface charge” interchangeable?

8) There are some strange phrases and mistakes, which should be corrected, e.g., “Recent years have seen..”.

Reviewer #3 (Remarks to the Author):

This manuscript reports the use of a branched polyethylimine to act as interface and stabilizing agent of small Au cluster on the surface of silica and other metal oxides. Due to the strong tendency of Au clusters to decompose and agglomerate forming Au nanoparticles, the potential use of these Au clusters is very limited. The authors address this issue of stability, proposing that the main mechanism of Au decomposition is oxidation of ligands and these could be avoided due to the antioxidant activity of the polyimine. In addition strong Coulombic forces maintain the Au cluster attached to the branched polymer onto the metal oxide surface, avoiding metal leaching. The authors have shown the advantage of the proposed methodology by using the Au cluster on silica coated and coated it with TiO₂, the resulting multicomponent nanoparticulated material acting as visible light photocatalyst for degradation of dyes and pollutants in water. Considering the interest in the use of Au clusters in catalysis and photocatalysis, the proposed methodology could be of large interest. Publication in Nature Communications is recommended with some

changes addressing the following comments:

- The presence of the TiO₂ shell increases considerably adsorption of the dye compared to silica. It is unclear how the authors have taken into account the increased adsorption when obtaining the corrected photocatalytic degradation data.
 - It is known that amorphous TiO₂ has much lower catalytic activity than crystalline TiO₂ phases. The present procedure does not allow the typical calcination at 350 °C to crystallize amorphous titania. A comment on this issue (and possibly a strategy to circumvent this issue) should be included in the manuscript.
 - To support the role of polyimine as stabilizer, the results of a control in which the SAP material is irradiated in the presence of an antioxidant will be important.
 - According to the author's proposal polyimine would undergo oxidation in the way to stabilize the Au cluster and would undergo presumably decomposition? Am I right? This indicates that the branched polyimine will be a sacrificial agent that eventually could lose activity. The authors should comment on this.
 - Similarly, the role of silica on the photocatalytic activity is unclear. Since the authors have shown that the use of branched polyimine serves for different metal oxides, would it be possible to use this approach to deposit Au clusters on anatase TiO₂ and test the photocatalytic activity?
- Other points:
- Which is the Au⁺/Au(0) proportion of the Au GSH clusters determined by XPS?
 - It is unclear why the authors use the prefix bio to indicate that the polyethylimine is reducible. Please clarify.
 - It is indicated in the text that Au clusters could be used as visible light photosensitizer. A comment on the λ_{max} absorption and other photophysical relevant properties of Au GSH as photosensitizer would be important.

Once these points have been addressed, publication in Nature Communications should proceed.

Dear Reviewers,

We greatly appreciate your insightful questions and comments which helped us significantly improve the quality of our manuscript. All of the questions and comments have been carefully addressed in the revised manuscript. The texts for Author reply are marked in blue and the texts for Corresponding revisions are marked in red in the following Point-by-Point response. For your convenience to quickly read the revised manuscript, the changes and additions are highlighted in red in the manuscript.

Thank you for your attention in your busy time !

Point-by-Point Response to Reviewers' Comments

Response to the comments of reviewer 1:

Comments:

Suppressing the aggregation of ultrasmall Au clusters during light irradiation is one of the important issues to be addressed in the application of Au clusters. This article reports a way of doing it by using poly-ethylenimine. At a quick glance, the role of this surface modification works very well and I think this strategy works to some extent.

Author reply:

We deeply appreciate the reviewer's agreement on this work. Your concerns have been carefully considered and we have revised our manuscript with great attention.

Comments 1: *However, the enhanced performance in photocatalysis is not necessarily from the result of better stability. The apparent difference in the interaction between SiO₂ and Au clusters must result in different loading of Au clusters on SiO₂. Therefore, a quantitative analysis for the loading amount should be performed and the photocatalytic activity must be normalized with respect to the determined amount of Au clusters. Otherwise, a fair comparison cannot be made.*

Author reply:

We greatly appreciate the reviewer's comment with regard to the loading amounts of Au GSH clusters in the SAB and SAP samples. We totally agree with the reviewer's opinion that the difference in the interaction between SiO₂ and Au GSH clusters will lead to the different loading amount of Au GSH clusters on SiO₂ surface. Accordingly, the loading amounts of Au GSH clusters in the SAB and SAP samples have been quantified by an inductively coupled plasma optical emission spectroscopy (ICP-OES). As shown in Supplementary Table 1, the addition amount of Au GSH clusters in SAB is demonstrated to be 0.86%, which is higher than that of SAP (0.28%). The different loading amounts of Au GSH clusters between SAB and SAP samples can be attributed to different interaction between SiO₂ and Au GSH clusters, as mentioned by the reviewer. The BPEI modified SiO₂ spheres for fabricating SAB sample exhibit relatively high positively charge surface with a zeta potential value of +36 mV (Supplementary Fig. 4a), which facilitates the strong electrostatic interaction between SiO₂ spheres with negatively charged Au GSH clusters (-21 mV, Supplementary Fig. 4c) via coulombic forces. In contrast, the SiO₂ spheres display a weak positive charge with a zeta potential value of +5 mV (Supplementary Fig. 4b) and the carboxylic groups in the glutathione ligand of Au GSH clusters are protonated under acidic conditions at pH 2 (**J. Am. Chem. Soc.** 2016, 138, 390-401), which subsequently leads to the inferior electrostatic attraction, resulting in the low addition amount of Au GSH clusters onto the surface of SiO₂ in SAP.

For a fair comparison on the photoactivity of various SABT and SAPT samples, the photocatalytic performances of SAPT composites have been normalized with respect to the loading amount of Au GSH clusters in SAB, as revealed in Supplementary Fig. 22 in the revised supplementary information. **The normalized RhB degradation efficiencies over different SAPT composites (Supplementary Fig. 22) are worse than that of SABT samples (Fig. 3a) under identical reaction conditions, which suggests that the presence of BPEI layer is beneficial for**

the photoactivity enhancement since the photostability of Au GSH clusters in the SAB samples is significantly ameliorated.

Supplementary Table 1. The loading amounts of Au GSH clusters in SAB and SAP samples quantified by ICP-OES.

Samples	SAB	SAP
Loading amount of Au GSH clusters	0.86%	0.28%

Supplementary Figure 4. The zeta potential values (ξ) of (a) BPEI-SiO₂ spheres; (b) SiO₂ sphere at pH 2 and (c) Au GSH clusters in water. The insets of a-c are the corresponding model illustrations.

Note: The ξ value of SiO₂ spheres at pH 2 is +5 mV. The ξ values of BPEI-SiO₂ spheres and Au GSH clusters in water without adjusting pH values are demonstrated to be +36 mV and -21 mV, respectively.

Supplementary Figure 22. Normalized photocatalytic degradation of RhB over blank SiO₂ spheres, SAP and SAPT composites with different TiO₂ shell thickness under visible light irradiation ($\lambda > 420$ nm) for 0.5 h.

Note: Due to the addition amount of Au GSH clusters in SAP is different from that of SAB

(Supplementary Table 1), the photocatalytic activity of RhB degradation over SAP, TAP and SAPT composites with different TiO₂ shell thickness has been normalized with respect to the loading amount of Au GSH clusters in SAB for a fair comparison.

Figure 3 | The enhancement of photocatalytic performance over SABL composites and the underlying mechanism. Photocatalytic degradation of (a) RhB over blank SiO₂ spheres, TiO₂, SAB, TAB and SABL composites with different TiO₂ shell thickness under visible light irradiation ($\lambda > 420$ nm) for 0.5 h; photocatalytic reduction of (b) *p*-methoxy nitrobenzene to *p*-methoxy aniline and over blank SiO₂ spheres, TiO₂, SAB, TAB and SABL composites with different TiO₂ shell thickness under visible light irradiation for 5 h; (c) recycling photocatalytic degradation of RhB over optimal SABL-0.15 composite under visible light irradiation ($\lambda > 420$ nm); (d) transient photocurrent densities of SiO₂ spheres, SAB and SABL composites with different TiO₂ shell thickness under visible light irradiation ($\lambda > 420$ nm); (e) scheme illustrating the photogenerated electron transport pathways between Au GSH clusters and TiO₂ shell; (f) cyclic voltammograms of SiO₂ spheres, SAB and SABL composites with different TiO₂ shell thickness; (g) bar plots showing the remaining RhB in reaction solutions after being kept in dark for 3 h to achieve the adsorption–desorption equilibrium over SiO₂ spheres, SAB and SABL composites with different TiO₂ shell thickness; (h) nitrogen adsorption–desorption isotherms of SiO₂ spheres, SAB and SABL composites with different TiO₂ shell thickness; (i) surface area of SiO₂ spheres, SAB and SABL composites with different TiO₂ shell thickness.

Corresponding revisions highlighted in red in the revised manuscript:

Lines 31-35 of Page 5 and Lines 1-4 of Page 6:

Notably, due to the protonation of carboxylic groups in the glutathione ligands under acidic conditions¹⁹ and the relatively weak electropositivity of SiO₂ spheres (Supplementary Fig. 4b), the as-prepared sample of SAP generally exhibits a weak Au–oxide interaction. The difference in the interaction between SiO₂ and Au GSH clusters over SAB and SAP samples may lead to the different loading of Au GSH clusters on SiO₂ surface, which has been investigated by the inductively coupled plasma emission spectroscopy. As revealed in Supplementary Table 1, the addition amount of Au GSH clusters in SAB (0.86%) is higher than that of SAP with 0.28% Au GSH clusters, which could be attributed to the strong coulombic force between positively charged SiO₂ spheres with BPEI modification and the negatively charged Au GSH clusters.

Lines 7-16 of Page 11:

Due to the addition amount of Au GSH clusters in SAP is different from that of SAB (Supplementary Table 1), the photocatalytic activities of RhB degradation over various SAPT composites with different TiO₂ shell thickness have been normalized with respect to the loading amount of Au GSH clusters in SAB for a fair comparison. As displayed in Supplementary Fig. 22, the coating of TiO₂ shell greatly improve the photocatalytic efficiency of SAP sample and the SAPT-0.15 composites exhibit the best photoactivity among these samples. However, the RhB degradation efficiency over SAPT-0.15 composites is worse than that of SABT-0.15 (Fig. 3a) under identical reaction conditions, which suggests that the presence of BPEI layer is beneficial for the photoactivity enhancement since the photostability of Au GSH clusters in the SABT-0.15 is significantly ameliorated.

Corresponding revisions highlighted in red in the revised supplementary information:

Supplementary Table 1 has been added, and **Supplementary Figure 4** and **22** have been revised in the revised supplementary information.

Comments 2: *And I am curious about the photostability of poly-ethylenimine. As stated in the manuscript, the authors claimed that BPEI can serve as a reducing agent to prevent the oxidation of the ligand of Au clusters, which implies that BPEI should be oxidized. This oxidative degradation is also anticipated given the decomposition of other organic substances demonstrated in this work. Hence, the protection capability of BPEI may be restricted if the irradiation is further prolonged. This possibility needs to be investigated.*

Author reply:

Thanks for your valuable comments. We certainly agree with the reviewer that BPEI may be consumed ultimately and the Au GSH clusters over SAB may lead to aggregation after BPEI layer is depleted if we irradiate the SAB sample for a very long time. In our original manuscript, the SAB sample has been irradiated under visible light irradiation ($\lambda > 420$ nm) for 24 h to investigate the effect of BPEI on inhibiting the fusion of Au GSH clusters. The results in Supplementary Fig. 8

suggest that the size of Au GSH clusters over SAB after 24 h visible light irradiation maintains unchanged, indicating that the BPEI layer as interfacial modification in the SAB system can provide a long time protection (24 h) with regard to the stabilization of Au GSH clusters under continuous visible light illumination ($\lambda > 420$ nm). **To address the reviewer's concern, the irradiation time over SAB has been further prolonged to 36 h and 48 h and the size information of the Au GSH clusters in SAB has also been studied.** As illustrated in Supplementary Fig. 9 in the revised supplementary information, after 36 h and 48 h visible light irradiation, the size of Au GSH clusters will increase to 2.0 nm and 2.1 nm, respectively. The slight increase of Au GSH clusters size may be attributed to the partial depletion of BPEI since the BPEI layer that serves as a reducing agent would undergo oxidation and/or decomposition in the way to stabilize the Au GSH clusters.

Particularly, we would like to emphasize that, despite various studies have observed the transformation of ultrasmall Au GSH clusters to larger Au NPs (**J. Am. Chem. Soc.** 2014, 136, 6075; **J. Phys. Chem. Lett.** 2013, 4, 2847; **ACS Appl. Mater. Interfaces** 2015, 7, 28105; **Sci. Rep.** 2016, 6, 22742), **the effective control of Au GSH clusters with long-term stability on the substrates under *in situ* photo-irradiation conditions still remains a challenge, which becomes the main bottleneck for the development of Au clusters-based catalysts systems for practical applications.** Our present work employs the BPEI as interfacial modification of SiO₂-Au GSH clusters composites, which can maintain the size and structure of Au GSH clusters over 24 h under continuous visible light irradiation ($\lambda > 420$ nm) on the surface of SiO₂ supports. This is the first research work regarding how to stabilize the ultrasmall Au GSH clusters for designing efficient and stable Au GSH clusters-based composite photocatalysts. Additionally, the surface of SAB composites has been subsequently coated by a thickness tunable TiO₂ shell for constructing core-shell SiO₂-Au GSH clusters-BPEI@TiO₂ (SABT) structures, which not only further contributes to stabilizing the ultrasmall Au GSH clusters but also improves the photocatalytic activities of Au GSH clusters during catalytic reactions under visible light illumination. Thus, our joint strategy *via* interfacial modification and sequential coating of semiconductor shell provides a simple and effective approach for stabilizing Au clusters with improved photocatalytic performance.

Supplementary Figure 8. TEM image (a) and HRTEM image (b) of SiO₂-Au GSH clusters-BPEI composites (SAB) after visible light irradiation ($\lambda > 420$ nm) for 24 h; size distribution histogram (c) of Au GSH clusters over SAB after visible light irradiation ($\lambda > 420$ nm) for 24 h.

Supplementary Figure 9. HRTEM images of SAB after visible light irradiation ($\lambda > 420$ nm) for (a) 36 h and (c) 48 h; size distribution histogram of Au GSH clusters over SAB after visible light irradiation ($\lambda > 420$ nm) for (b) 36 h and (d) 48 h; (e) EDX spectrum of SAB originated from **Supplementary Fig. 9c**.

Note: The EDX spectrum in **Supplementary Fig. 9e** evidences the presence of Au, O and Si elements over SAB sample and the detected element Cu can be attributed to the use of Cu grid, which serves as the support for TEM analysis.

Corresponding revisions highlighted in red in the revised manuscript:

Lines 27-35 of Page 6 and Lines 1-2 of Page 7:

When the irradiation time over SAB is further prolonged to 36 h and 48 h, the size of Au GSH clusters will increase to 2.0 nm and 2.1 nm, respectively, as illustrated in Supplementary Fig. 9. The slight increase of Au GSH clusters size is reasonable, which may be attributed to the partial depletion of BPEI since the BPEI layer that serves as a reducing agent would undergo oxidation and/or decomposition in the way to stabilize the Au GSH clusters.

Corresponding revisions highlighted in red in the revised supplementary information:

Supplementary Figure 9 has been added in the revised supplementary information.

Comments 3: *SABT-0.15 is claimed as the best catalyst among other counterparts for the decomposition of RhB and the reduction of Cr(IV) and p-methoxy nitrobenzene. However, the results from the PEC measurements don't comply with this trend. SABT-0.05 works best, which is attributed to the thin TiO₂ layer. With the thicker TiO₂ coating, the catalytic activity get lower due to the increased recombination. These observations are somewhat confusing. Both experimental situations are different, but in terms of charge recombination they share the same recombination dynamics.*

Author reply:

Thanks for your comments. **It has been well documented that the performance of photocatalysts is determined not only by the charge carrier separation efficiency but also the adsorption ability of catalysts (J. Phys. Chem. C 2011, 115, 14300–14308; J. Mater. Chem. 2012, 22, 1160–1166; J. Phys. Chem. C 2016, 120, 265–273).**

In this work, the photoelectrochemical (PEC) performances in Fig. 3d and f suggest that the separation efficiency of photoinduced charge carriers over the SABT-0.05 composite with thin layer of TiO₂ is the best among these samples because a thicker TiO₂ shell will block the transport of photogenerated electrons from Au GSH clusters to the back contact, thus reducing the overall PEC performances. Notably, with the increased thickness of TiO₂ shell, the adsorption capacity of SABT composites gradually increases, as shown in Fig. 3g, and the SABT-0.2 exhibits the strongest adsorption toward RhB among these samples. **Therefore, the dye adsorption and charge carrier separation should have a synergistic effect on the catalytic performance, which rationalizes the best photoactivity for dye degradation achieved neither over SABT-0.2 with the highest RhB adsorption capacity nor SABT-0.05 with the most efficient charge carrier separation efficiency. It is the SABT-0.15, which could balance the combined influence of the RhB adsorption and charge carrier separation, that acquires the best photocatalytic performance.** Based on the above discussion, the primary role of coating TiO₂ semiconductor layer onto SAB for ameliorating the photocatalytic efficiency of SABT is two-fold. **One is to improve the separation and migration efficiency of photoinduced charge carriers from excited Au GSH clusters. The other role of TiO₂ shell is to enhance the adsorption capacity of the catalysts toward the reactants.** By adjusting the thickness of TiO₂ shells, the photoactivity of SABT can be tuned due to cooperative adsorption and PEC properties. Therefore, the optimal SABT-0.15 composite that exhibits the highest activity toward photocatalytic degradation of RhB among these SABT samples is reasonable and understandable.

Figure 3 | The enhancement of photocatalytic performance over SABT composites and the underlying mechanism. Photocatalytic degradation of (a) RhB over blank SiO₂ spheres, TiO₂, SAB, TAB and SABT composites with different TiO₂ shell thickness under visible light irradiation ($\lambda > 420$ nm) for 0.5 h; photocatalytic reduction of (b) *p*-methoxy nitrobenzene to *p*-methoxy aniline and over blank SiO₂ spheres, TiO₂, SAB, TAB and SABT composites with different TiO₂ shell thickness under visible light irradiation for 5 h; (c) recycling photocatalytic degradation of RhB over optimal SABT-0.15 composite under visible light irradiation ($\lambda > 420$ nm); (d) transient photocurrent densities of SiO₂ spheres, SAB and SABT composites with different TiO₂ shell thickness under visible light irradiation ($\lambda > 420$ nm); (e) scheme illustrating the photogenerated electron transport pathways between Au GSH clusters and TiO₂ shell; (f) cyclic voltammograms of SiO₂ spheres, SAB and SABT composites with different TiO₂ shell thickness; (g) bar plots showing the remaining RhB in reaction solutions after being kept in dark for 3 h to achieve the adsorption–desorption equilibrium over SiO₂ spheres, SAB and SABT composites with different TiO₂ shell thickness; (h) nitrogen adsorption–desorption isotherms of SiO₂ spheres, SAB and SABT composites with different TiO₂ shell thickness; (i) surface area of SiO₂ spheres, SAB and SABT composites with different TiO₂ shell thickness.

Corresponding revisions highlighted in red in the revised manuscript:

Figure 3 has been revised in the revised manuscript.

Comments 4: *This work looks systematically performed and the analyses are quite thorough. The idea of using BPEI is interesting, but I do not think the significance of this work meets the requirement for the publication in this journal. Scientific Reports or other relevant journals would be a better place for this work if all the concerns are properly addressed.*

Author reply:

At first, we appreciate the reviewer's evaluation that our work is interesting and the analyses are quite thorough. However, **we do not think the assessment that the significance of this work cannot meet the requirement of the journal of Nature Communications is objective**, in view of the following background in this significant research field.

Noble metal nanoparticles (MNPs) are uniquely suited for heterogeneous catalysis because of their relatively high specific surface areas and abundant active centers. The size of MNPs has been shown to be one of the most important factors that dictates the performance of a catalyst. Recently, considerable interest has been shown in controlling ultrasmall gold (Au) nanoparticles with atomic precision, which are often called Au clusters, due to their distinctive properties. These ligands protected Au clusters, typically glutathione-protected Au clusters (Au GSH clusters), can serve as both photosensitizer and catalytic center for multifunctional use in photoredox catalysis, such as water splitting, selective organic synthesis and pollutants degradation (for example, see: **Chem. Rev.** 2016, 116, 10346-10413; **Acc. Chem. Res.** 2013, 46, 1749-1758; **Adv. Mater.** 2010, 22, 3185-3188; **J. Am. Chem. Soc.** 2016, 138, 390-401; **J. Am. Chem. Soc.** 2013, 135, 8822-8825). **However, such ultrasmall Au GSH clusters suffer from serious instability under light irradiation due to its extremely high surface energy and large surface.** The loading of Au GSH clusters onto different supports is often inefficient to avoid coalescence and agglomeration of these Au GSH clusters. **Actually, up to now, the efficient control of long-term photostability of ultrasmall Au GSH clusters on the metal-support interface has never been achieved.**

In this paper, we report a facile and general strategy for enhancing the photostability of Au GSH clusters loaded on the surface of SiO₂ sphere by utilizing multifunctional branched poly-ethylenimine (BPEI) as a surface charge modifying, reducing and stabilizing agent. In addition, while simultaneously maintaining the photostability of Au GSH clusters, **the sequential coating of TiO₂ shells for constructing core-shell SiO₂-Au GSH clusters-BPEI@TiO₂ (SABT) nanostructures has been demonstrated to significantly ameliorate the photoactivity by regulating the photoelectrochemical and adsorption properties of SABT composites synergistically. This work is expected to stimulate broad scientific and technological interests in this special type of metal nanomaterial and provide a guideline in structural design of Au clusters-based composites with improved catalytic performance and long-term photostability.**

We believe that this manuscript is of high interest to the general readership of *Nature Communications* for the following two fundamental reasons:

Firstly, regarding the reported Au GSH clusters-based composite photocatalysts, the aggregation of Au GSH clusters at the metal/semiconductor interface under light illumination has been widely observed (**J. Am. Chem. Soc.** 2014, 136, 6075-6082; **J. Phys. Chem. Lett.** 2013, 4, 2847-2852; **ACS Appl. Mater. Interfaces** 2015, 7, 28105-28109; **Sci. Rep.** 2016, 6, 22742). **However, the effective control of these ultrasmall Au GSH clusters with long-term stability on the substrates under in situ photo-irradiation conditions still remains a challenge**, which becomes the main bottleneck for the development of Au clusters-based catalysts systems for the long-term use. In this

work, we report a facile and general strategy of using multifunctional branched poly-ethylenimine as surface charge modifying, reducing and stabilizing agents to enhance the photostability of Au GSH clusters loaded on the surface of SiO₂ sphere supports.

Secondly, although some excellent works of Au clusters-semiconductor composites have been reported for photocatalytic solar energy conversions by randomly decorating Au clusters onto the surface of semiconductor supports, the rational structure design for achieving high efficient Au clusters-semiconductor composites is barely explored in previous reports. In this work, while simultaneously maintaining the photostability of Au GSH clusters, **core-shell SiO₂-Au GSH clusters-BPEI@TiO₂ (SABT) composites have been constructed by coating of the thickness-controlled TiO₂ shells to significantly ameliorate the photoactivity.** Besides, the adsorption and photoelectrochemical properties of SABT composites can be well regulated by simply adjusting the thickness of TiO₂ shell, thereby synergistically ameliorating the photoactivity of SABT composites.

It is anticipated that this work could raise broad scientific and technological interests in this special type of metal nanomaterial and offer an avenue for rational designing of Au clusters-based composites photocatalyst with high catalytic efficiency and long-term stability for photoredox catalysis.

Thank you very much for your positive, valuable and constructive comments to help us improve the quality of the manuscript with great attention!

Response to the comments of reviewer 2:

Comments:

The manuscript on preparation of highly stable gold nanoclusters by B. Weng et al. is very interesting. Although, the overall quality of the manuscript is high, there are some important issues that should be explained before possible publications, as shown below:

Author reply:

We deeply appreciate the reviewer's evaluation that our work is very interesting and the overall quality of the manuscript is high. We have revised our manuscript with great attention according to your valuable comments.

Comments 1: *The mechanism must be clarified. Authors proposed photo-generation of electrons (able to form reactive oxygen species) and high stability of the system. This two contradict propositions must be explained, i.e., gold should be oxidized by this mechanism, and thus either it can be dissolved or electron donor should be applied (considering glutathione as electron donor, its regeneration should be proposed),*

Author reply:

Thanks for your valuable comments. We agree with the reviewer's opinion that the presence of reactive oxygen species (ROSs) is unfavorable for the stability of Au GSH clusters and it has been reported that the onslaught of ROSs on organic ligands of Au GSH clusters can lead to the instability of Au GSH clusters and their aggregation into large Au nanoparticles (**ACS Appl. Mater. Interfaces** 2015, 7, 28105-28109; **Sci. Rep.** 2016, 6, 22742). Notably, in our reaction systems, the Au GSH clusters is found to be stable during photocatalytic reactions and the following factors are considered to be the decisive role for achieving high photostability of Au GSH clusters in SABT composites.

On one hand, BPEI with high reductive capability could serve as an electron donor and/or an effective reducing and stabilizing agent to protect the organic ligands of Au GSH clusters from being oxidized by ROSs during the photo-irradiation process. Specifically, due to the presence of BPEI layer, the R-SO₃ species belonging to the oxidation products of glutathione have not been detected even after a long time irradiation (24 h) over the SAB composites, which indicates the critical role of BPEI layer in stabilizing Au GSH clusters against aggregation under continuous light illumination condition. Furthermore, the multiple polymer chains of primary, secondary and tertiary amine groups in BPEI could encapsulate the as-synthesized Au GSH clusters via crosslinking (**Chem. Commun.**, 2014, 50, 88-90), which could fix these clusters on the surface of supports and hamper the migration of clusters for fusing, thus stabilizing the ultrasmall Au GSH clusters.

On the other hand, the intelligent design of core-shell SABT composites by coating TiO₂ shell can enhance the adsorption capacity of catalysts toward reactants (Fig. 3g), which would consume the ROSs timely and effectively to reduce the possibility of the reactions between ROSs with the organic ligands of Au GSH clusters, thus leading to the improvement of the photostability of Au GSH clusters in SABT composites.

Figure 3 | The enhancement of photocatalytic performance over SABT composites and the underlying mechanism. Photocatalytic degradation of (a) RhB over blank SiO₂ spheres, TiO₂, SAB, TAB and SABT composites with different TiO₂ shell thickness under visible light irradiation ($\lambda > 420$ nm) for 0.5 h; photocatalytic reduction of (b) *p*-methoxy nitrobenzene to *p*-methoxy aniline and over blank SiO₂ spheres, TiO₂, SAB, TAB and SABT composites with different TiO₂ shell thickness under visible light irradiation for 5 h; (c) recycling photocatalytic degradation of RhB over optimal SABT-0.15 composite under visible light irradiation ($\lambda > 420$ nm); (d) transient photocurrent densities of SiO₂ spheres, SAB and SABT composites with different TiO₂ shell thickness under visible light irradiation ($\lambda > 420$ nm); (e) scheme illustrating the photogenerated electron transport pathways between Au GSH clusters and TiO₂ shell; (f) cyclic voltammograms of SiO₂ spheres, SAB and SABT composites with different TiO₂ shell thickness; (g) bar plots showing the remaining RhB in reaction solutions after being kept in dark for 3 h to achieve the adsorption–desorption equilibrium over SiO₂ spheres, SAB and SABT composites with different TiO₂ shell thickness; (h) nitrogen adsorption–desorption isotherms of SiO₂ spheres, SAB and SABT composites with different TiO₂ shell thickness; (i) surface area of SiO₂ spheres, SAB and SABT composites with different TiO₂ shell thickness.

Comments 2: *Dyes cannot be used for activity testing due to sensitization mechanisms (see reviews by B. Ohtani), other colorless compounds should be used for mechanism study (Application of chromate as testing molecule is also not recommended since it could sensitize titania (as has been published, e.g., Kuncewicz et al.)).*

Author reply:

Thanks for your helpful comments. We agree with the reviewer that the dye and chromate sensitized mechanisms could be present in the photocatalytic activity test. **Actually, to eliminate possible dyes (*i.e.*, RhB, and MB) and chromate (*i.e.*, Cr(VI)) photosensitization effect on the activity, photocatalytic reduction of *p*-methoxy nitrobenzene to *p*-methoxy aniline over various SABT composites has also been performed in the original manuscript under visible light irradiation ($\lambda > 420$ nm), as shown in Fig. 3b in the revised manuscript (*i.e.*, Fig. 16 in the original Supplementary Information). The higher photocatalytic activity of SABT composites than that of SiO₂ spheres and SAB sample verifies the positive role of TiO₂ shell in improving the photoactivity of SiO₂-Au GSH clusters composites. And the SABT-0.15 shows the best photoactivity among these samples, elucidating that the thickness of TiO₂ shell plays a critical role in determining the photocatalytic performance of SABT toward *p*-methoxy nitrobenzene reduction. Moreover, due to the intelligently designed core-shell structure, all of the SABT composites show higher catalytic activity than the TiO₂-Au GSH cluster-BPEI (TAB) composite. **The photoactivity tendency toward *p*-methoxy nitrobenzene reduction over SABT composites is similar with that of photocatalytic degradation of dyes and reduction of Cr(VI) and further corroborates our conclusions.****

Additionally, it is well known that the sensitization efficiency of RhB during the photocatalytic reaction is generally low (*Appl. Catal. B* 2007, 69, 138–144; *Chem. Commun.* 2014, 50, 6637–6640; *Nanoscale* 2015, 7, 7030–7034). As displayed in Fig. 3, both bare TiO₂ and SiO₂ show negligible photoactivity toward RhB degradation, indicating that the photoactive ingredients of various SABT samples are the Au GSH clusters. Anyway, as the reviewer recommended, the photocatalytic performances of MB degradation and Cr(VI) reduction over these composites have been removed based on the researches from B. Ohtani (*Catalysts*, 2016, 6, 192) and Kunczewicz (*Catal. Today*, 2011, 161, 78-83).

Figure 3 | The enhancement of photocatalytic performance over SABT composites and the underlying mechanism. Photocatalytic degradation of (a) RhB over blank SiO₂ spheres, TiO₂, SAB, TAB and SABT composites with different TiO₂ shell thickness under visible light irradiation ($\lambda > 420$ nm) for 0.5 h; photocatalytic reduction of (b) *p*-methoxy nitrobenzene to *p*-methoxy aniline and over blank SiO₂ spheres, TiO₂, SAB, TAB and SABT composites with different TiO₂ shell thickness under visible light irradiation for 5 h; (c) recycling photocatalytic degradation of RhB over optimal SABT-0.15 composite under visible light irradiation ($\lambda > 420$ nm); (d) transient photocurrent densities of SiO₂ spheres, SAB and SABT composites with different TiO₂ shell thickness under visible light irradiation ($\lambda > 420$ nm); (e) scheme illustrating the photogenerated electron transport pathways between Au GSH clusters and TiO₂ shell; (f) cyclic voltammograms of SiO₂ spheres, SAB and SABT composites with different TiO₂ shell thickness; (g) bar plots showing the remaining RhB in reaction solutions after being kept in dark for 3 h to achieve the adsorption–desorption equilibrium over SiO₂ spheres, SAB and SABT composites with different TiO₂ shell thickness; (h) nitrogen adsorption–desorption isotherms of SiO₂ spheres, SAB and SABT composites with different TiO₂ shell thickness; (i) surface area of SiO₂ spheres, SAB and SABT composites with different TiO₂ shell thickness.

Corresponding revisions highlighted in red in the revised manuscript:

Figure 3 has been revised in the revised manuscript.

Lines 18-30 of Page 11:

To investigate the effect of surface coating of TiO₂ shell on the photocatalytic performance and exclude the possible dye photosensitization effect of RhB on the activity⁵⁶, the visible light photoactivities of SiO₂ spheres, TAB, SAB and SABT composites towards photocatalytic reduction of *p*-methoxy nitrobenzene to *p*-methoxy aniline have been evaluated, which is a typical six-electron reduction reaction (Supplementary Equation 1)^{24, 27, 47, 57}. As presented in Fig. 3b, the higher photocatalytic activity of SABT composites than that of SiO₂ spheres and SAB sample verifies the positive role of TiO₂ shell in improving the photoactivity of SiO₂-Au GSH clusters composites. And the SABT-0.15 shows the best photoactivity among these samples, elucidating that the thickness of TiO₂ shell plays a critical role in determining the photocatalytic performance of SABT toward *p*-methoxy nitrobenzene reduction. Moreover, due to the intelligently designed core-shell structure, all of the SABT composites show higher catalytic activity than the TAB composite.

Comments 3: *Authors concluded that titania played important role in the mechanism (this is obvious), but they did not perform any reference experiments for: (i) bare titania and (ii) titania with gold nanoclusters.*

Author reply:

We deeply thank the reviewer for this valuable comment. As the reviewer recommended, the photoactivities of bare TiO₂ and TiO₂-Au GSH clusters-BPEI composite (TAB) toward photocatalytic RhB degradation and reduction of *p*-methoxy nitrobenzene have been supplemented in our revised manuscript. As illustrated in Fig. 3, it is clear that bare TiO₂ catalyst exhibits negligible photoactivity since the visible light ($\lambda > 420$ nm) cannot excite semiconductor TiO₂ with a band gap of 3.2 eV to generate electron-hole pairs for driving the photocatalytic reactions. Notably, the TAB sample with the decoration of Au GSH clusters shows obvious photoactivity toward degradation of RhBs and reduction of *p*-methoxy nitrobenzene, which confirms that the Au GSH clusters can act as photosensitizers to generate electron-hole pairs under visible light irradiation, thereby triggering various photocatalytic reactions. Furthermore, all of SABT composites exhibit higher photocatalytic efficiency than TAB sample, which indicates that the intelligently designed core-shell structure of SABT is favorable for enhancing the photocatalytic performance of Au GSH clusters-semiconductor composites.

Figure 3 | The enhancement of photocatalytic performance over SABT composites and the underlying mechanism. Photocatalytic degradation of (a) RhB over blank SiO₂ spheres, TiO₂, SAB, TAB and SABT composites with different TiO₂ shell thickness under visible light irradiation ($\lambda > 420$ nm) for 0.5 h; photocatalytic reduction of (b) *p*-methoxy nitrobenzene to *p*-methoxy aniline and over blank SiO₂ spheres, TiO₂, SAB, TAB and SABT composites with different TiO₂ shell thickness under visible light irradiation for 5 h; (c) recycling photocatalytic degradation of RhB over optimal SABT-0.15 composite under visible light irradiation ($\lambda > 420$ nm); (d) transient photocurrent densities of SiO₂ spheres, SAB and SABT composites with different TiO₂ shell thickness under visible light irradiation ($\lambda > 420$ nm); (e) scheme illustrating the photogenerated electron transport pathways between Au GSH clusters and TiO₂ shell; (f) cyclic voltammograms of SiO₂ spheres, SAB and SABT composites with different TiO₂ shell thickness; (g) bar plots showing the remaining RhB in reaction solutions after being kept in dark for 3 h to achieve the adsorption–desorption equilibrium over SiO₂ spheres, SAB and SABT composites with different TiO₂ shell thickness; (h) nitrogen adsorption–desorption isotherms of SiO₂ spheres, SAB and SABT composites with different TiO₂ shell thickness; (i) surface area of SiO₂ spheres, SAB and SABT composites with different TiO₂ shell thickness.

Corresponding revisions highlighted in red in the revised manuscript:

Lines 34-35 of Page 10 and Lines 1-6 of Page 11:

Furthermore, the sample of TiO₂-Au GSH cluster-BPEI (TAB) exhibits worse catalytic activity than

the SABT composites, which indicates that the intelligently designed core-shell structure of SABT is favorable for enhancing the photocatalytic performance of Au GSH clusters-semiconductor composites. The pseudo-first order kinetic of the degradation of RhB based on the above data is shown in Supplementary Table 2. It is clear that all of the SABT samples show higher reaction rate than the SAB and TAB samples, among which SABT-0.15 exhibits the highest reaction rate of 0.119 min^{-1} among these SABT composites.

Page 26:

The photoactivities of bare TiO_2 and TiO_2 -Au GSH clusters-BPEI (TAB) have been supplemented in the revised **Figure 3**.

Comments 4: *What about photocatalytic activity of ZnO-Au GSH clusters-BPEI (shown in Supplementary Figure 9)?*

Author reply:

Thanks for your kind comments. To address the reviewer's concern, the photocatalytic activity of ZnO-Au GSH clusters-BPEI (ZAB) has been evaluated toward RhB degradation under visible light illumination ($\lambda > 420 \text{ nm}$) for 9 h. As shown in Supplementary Figure 18a, nearly 96% of RhB can be degraded over ZAB. To investigate the dye sensitization effect of RhB on the activity, the photocatalytic performance of bare ZnO has also been tested. Notably, the sample of bare ZnO removes ca.79% of RhB with 9 h of visible light irradiation, which is governed by a dye self-sensitization process because the wide-band gap of ZnO semiconductor (3.2 eV) cannot be activated in the visible light region (*J. Photochem. Photobiol. A* 2004, 162, 317–322; *J. Mater. Chem. A* 2014, 2, 9380-9389). The ZAB with the decoration of Au GSH clusters exhibits moderate photoactivity enhancement than bare ZnO, which suggests that the random loading of ultrasmall Au GSH clusters onto ZnO supports without rational structure design is inefficient to achieve high efficient Au GSH clusters-semiconductor composites. For your kind reference, the photocatalytic activities toward RhB degradation over ZAB composite and ZnO sample have been displayed as follows.

Supplementary Figure 18. Photodegradation of RhB over (a) bare ZnO and ZnO-Au GSH clusters-BPEI composites (ZAB) for 9 h and (b) rutile TiO_2 and rutile TiO_2 -Au GSH clusters-BPEI composites (RTAB) for 6 h under visible light irradiation ($\lambda > 420 \text{ nm}$).

Corresponding revisions highlighted in red in the revised manuscript:

Lines 11-23 of Page 9:

Notably, it has been demonstrated that the CB edge potential of metal oxides (*e.g.*, ZnO and rutile TiO₂) is more positive than the LUMO potential of Au GSH clusters, which enables the transformation of photoexcited electrons from Au GSH clusters to the metal oxide supports^{15, 24, 53}. The photoactivities of BPEI modified metal oxides-Au GSH clusters composites (denoted as MAB) have been evaluated toward RhB degradation under visible light irradiation ($\lambda > 420$ nm). As shown in Supplementary Fig. 18, the samples of MAB exhibit moderate photoactivity enhancement than bare semiconductors, which suggests that the random loading of ultrasmall Au GSH clusters onto semiconductors without rational structure design is inefficient to achieve high efficient Au GSH clusters-semiconductor composites. Therefore, a thickness tunable TiO₂ shell has been coated onto the surface of SAB composites for designing core-shell SiO₂-Au GSH clusters-BPEI@TiO₂ structures to construct high performance Au GSH clusters-semiconductor composites for solar energy conversion.

Page 22, References

Ref. 53 has been added.

53. Negishi Y. Toward the creation of functionalized metal nanoclusters and highly active photocatalytic materials using thiolate-protected magic gold clusters. *Bull. Chem. Soc. Jpn.* **87**, 375-389 (2013).

Corresponding revisions highlighted in red in the revised supplementary information:

Supplementary Figure 18 has been added.

Comments 5: *Although, the existence of optimal thickness of titania is not surprising, especially in the case of amorphous titania with plenty of recombination sites (and many reports on similar behaviors have been already published; some of them should be cited), there is a question why crystallization of titania was not performed.*

Author reply:

Thanks for your valuable comments. We agree with the reviewer's opinion that the amorphous TiO₂ with defects could exhibit worse photoactivity than crystalline TiO₂, which has also been demonstrated by previous works (**J. Phys. Chem. B**, 1997, 101, 3746-3752; **Appl. Catal. B**, 2000, 26, 207-215; **J. Phys. Chem. C**, 2008, 112, 3083-3089; **J. Mater. Chem.**, 2006, 16, 77-82; **Energy Environ. Sci.**, 2015, 8, 286-296). According to your precious suggestion, we have paid our endeavors to enhance the crystallization of TiO₂, including calcination and hydrothermal treatment, since the high temperature is generally required to transform amorphous TiO₂ particle into an anatase one.

The sample of SABT-0.15 has been calcinated at 350 °C under Ar for 2 h and the calcinated sample is denoted as SABT-0.15-350. Even though the peak of anatase TiO₂ (A-TiO₂) can be observed in the XRD pattern (Supplementary Fig. 23c), the color of SABT-0.15-350 sample changes to purple, as shown in Supplementary Fig. 23b, which corresponds to the color of Au nanoparticles (NPs), indicating that the Au GSH clusters aggregate into large Au NPs due to the high calcination

temperature. Furthermore, the DRS spectrum of SABB-0.15-350 sample in Supplementary Fig. 23a exhibit a surface plasmon resonance (SPR) peak located at 550 nm belonging to the metallic Au NPs, confirming the fusion of Au GSH clusters into Au NPs with large size. More direct evidence comes from the TEM image of SABB-0.15-350 sample, as pictured in Supplementary Fig. 23d, and the size of Au NPs is demonstrated to be 7.8 nm (Supplementary Fig. 23e), which suggests that the calcination is unsuitable for enhancing the crystallinity of TiO₂ since the high temperature can lead to the aggregation of Au GSH clusters.

We then treat the SABB-0.15 composite under hydrothermal condition at 180 °C for 12 h for crystallizing TiO₂, and the obtained sample is labeled as SABB-0.15-180. The application of elevated temperatures and pressures in an aqueous solution could facilitate the conversion of amorphous TiO₂ into crystalline TiO₂ and cause an increase in its crystallinity. The XRD result in Supplementary Fig. 24c indicates that the TiO₂ in SABB-0.15-180 sample is anatase (A-TiO₂). However, the intelligently designed core-shell structure of SABB-0.15 is destroyed during the hydrothermal process, as revealed by TEM image in Supplementary Fig. 24d. Moreover, the coalescence of Au GSH clusters into larger metallic Au NPs has also been observed and confirmed by a series of techniques, as displayed in Supplementary Fig. 24a, b and d. The size of Au NPs in SABB-0.15-180 sample is calculated to be 3.0 nm (Supplementary Fig. 24e). The fusion of Au GSH clusters may be ascribed to the high temperatures and pressures during the hydrothermal process, which is essential for crystallizing TiO₂.

The photocatalytic performances of SABB-0.15-350 and SABB-0.15-180 composites have been evaluated toward photocatalytic RhB degradation and reduction of *p*-methoxy nitrobenzene under visible light illumination ($\lambda > 420$ nm), as shown in Supplementary Fig. 25. Both the SABB-0.15-350 and SABB-0.15-180 samples exhibit poor photoactivity as compared with SABB-0.15 composite, which could be attributed to the aggregation of Au GSH clusters into large metallic Au NPs and the destroy of core-shell structure of SABB-0.15-180 composite. According to your precious suggestion, these results have been supplemented in our revised manuscript to provide more information for the readership with citation of relevant references in our revised manuscript.

Supplementary Figure 23. (a) DRS spectrum; (b) digital photograph; (c) XRD pattern; (d) HRTEM image of SABT-0.15 composite calcinated at 350 °C under Ar for 2 h (denoted as SABT-0.15-350); (e) size distribution histogram of Au nanoparticles over SABT-0.15-350 sample; (f) EDX spectrum of SABT-0.15-350 originated from **Supplementary Fig. 23d**.

Supplementary Figure 24. (a) DRS spectrum; (b) digital photograph; (c) XRD pattern; (d) HRTEM image of SABT-0.15 composite calcinated at 180 °C for 12 h (denoted as SABT-0.15-180); (e) size distribution histogram of Au nanoparticles over SABT-0.15-180 sample; (f) EDX spectrum of SABT-0.15-180 originated from **Supplementary Fig. 24d**.

Supplementary Figure 25. Photocatalytic degradation of (a) RhB over SBT-0.15, SBT-0.15-350 and SBT-0.15-180 composites under visible light irradiation ($\lambda > 420$ nm) for 0.5 h; photocatalytic reduction of (b) *p*-methoxy nitrobenzene to *p*-methoxy aniline over SBT-0.15, SBT-0.15-350 and SBT-0.15-180 composites under visible light irradiation ($\lambda > 420$ nm) for 5 h.

Corresponding revisions highlighted in red in the revised manuscript:

Lines 30-35 of Page 11 and Lines 1-4 of Page 12:

In addition to the TiO₂ shell thickness, the crystallization of TiO₂ layer in composites is also a factor that affects the photocatalytic performance of SBT composites and it has been reported that the amorphous TiO₂ usually exhibits poor photocatalytic activity⁵⁸⁻⁶². We have paid our endeavors to crystallize TiO₂ shell in SBT-0.15, including calcination and hydrothermal treatment, since the high temperature is generally required to transform amorphous TiO₂ particle into an anatase one. Even though the crystallization of TiO₂ can be improved by calcination and hydrothermal treatment, the Au GSH clusters in SBT-0.15 suffer from serious instability under high temperature and aggregate into metallic Au nanoparticles with large size, as displayed in Supplementary Fig. 23-24, thereby deteriorating the photocatalytic performance of SBT-0.15 samples toward various photocatalytic reactions under visible light illumination ($\lambda > 420$ nm) (Supplementary Fig. 25).

Page 23, References

Ref. 58-62 have been added.

58. Sakatani Y, Grosso D, Nicole L, Boissiere C, de A. A. Soler-Illia GJ, Sanchez C. Optimised photocatalytic activity of grid-like mesoporous TiO₂ films: effect of crystallinity, pore size distribution, and pore accessibility. *J. Mater. Chem.* **16**, 77-82 (2006).
59. Liu H, Joo JB, Dahl M, Fu L, Zeng Z, Yin Y. Crystallinity control of TiO₂ hollow shells through resin-protected calcination for enhanced photocatalytic activity. *Energy Environ. Sci.* **8**, 286-296 (2015).
60. Zhang Q, Gao L, Guo J. Effects of calcination on the photocatalytic properties of nanosized TiO₂ powders prepared by TiCl₄ hydrolysis. *Appl. Catal., B* **26**, 207-215 (2000).
61. Ohtani B, Ogawa Y, Nishimoto S-i. Photocatalytic activity of amorphous-anatase mixture of titanium(IV) oxide particles suspended in aqueous solutions. *J. Phys. Chem. B* **101**, 3746-3752 (1997).

62. Tian G, Fu H, Jing L, Xin B, Pan K. Preparation and characterization of stable biphasic TiO₂ photocatalyst with high crystallinity, large surface area, and enhanced photoactivity. *J. Phys. Chem. C* **112**, 3083-3089 (2008).

Corresponding revisions highlighted in red in the revised supplementary information:

Supplementary Figures 22-24 have been added and the corresponding discussions have also been supplemented in the revised Supplementary Information.

Comments 6: *Authors presented plenty of supporting data, but some of them were not discussed, e.g., interesting data shown in Supplementary Figure 22 (impedance spectroscopy) should be discussed (at least in supplementary material file) and values on Y-axis should be shown.*

Author reply:

Thanks for your kind comments. As you kindly suggested, the discussion of the impedance spectroscopy in Supplementary Fig. 28 (Supplementary Fig. 22 in the original manuscript) has been supplemented in the revised supplementary information and the figure of impedance spectroscopy has also been revised to show the values on Y-axis. Furthermore, more discussions of data in the revised supplementary information have been added for the beneficial of readership.

Corresponding revisions highlighted in red in the revised supplementary information:

The discussions on **Supplementary Figures 4, 9-10, 21-24, and 28** have been added in the revised supplementary information.

Comments 7: *Do authors use “Zeta potential” and “surface charge” interchangeable?*

Author reply:

Thanks for your kind comments. As is well known, zeta potential analysis provides an insight into the characteristics of a sample. **Surface charge and zeta potential are not the same but they are related.** The zeta potential is the potential difference between bulk solution and the “shear plane” (particle surface + 1st ion layer [Helmholtz rigid double layer] + diffuse layer [Gouy-Chapman double layer] = electrically neutral). Particle movement in the solution leads to shearing (friction) of some particles on the outer limit of the diffuse layer. Such particles are no longer electrically neutral and the potential is called zeta potential. The zeta potential is a (relative) indicator for the surface potential of the particle and has been widely applied for providing the surface electronic properties of particle surface (*Nat. Commun.* 2017, 8, 14224; *J. Am. Chem. Soc.* 2015, 137, 2844-2847; *Nano Energy* 2015, 13, 757-770; *J. Catal.* 2015, 330, 387-395; *Chem. Commun.* 2015, 51, 7176-7179; *J. Mater. Chem. A* 2015, 3, 18622-18635; *Appl. Catal. B* 2016, 192, 8-16; *Nanoscale* 2014, 6, 6335-6345).

Comments 8: *There are some strange phrases and mistakes, which should be corrected, e.g., “Recent years have seen..”.*

Author reply:

Thanks for your kind comments. As you kindly suggested, the sentence of “Recent years have seen a surge of interest in loading ligand-protected gold (Au) clusters as visible light photosensitizers onto various supports for photoredox catalysis” has been revised as “Recently, loading ligand-protected gold (Au) clusters as visible light photosensitizers onto various supports for photoredox catalysis have attracted considerable attention”. Additionally, we have checked the whole article with great attention for correcting the mistakes to further improve the quality of our manuscript.

Corresponding revisions highlighted in red in the revised manuscript:

Page 2, Abstract:

Recently, loading ligand-protected gold (Au) clusters as visible light photosensitizers onto various supports for photoredox catalysis have attracted considerable attention.

Several grammar mistakes in **Page 8-10, 12 and 13** have been corrected and marked in red font for improving the quality of the manuscript.

Thank you very much for your positive, valuable and constructive comments to help us improve the quality of the manuscript with great attention!

Response to the comments of reviewer 3:

Comments:

This manuscript reports the use of a branched polyethylimine to act as interface and stabilizing agent of small Au cluster on the surface of silica and other metal oxides. Due to the strong tendency of Au clusters to decompose and agglomerate forming Au nanoparticles, the potential use of these Au clusters is very limited. The authors address this issue of stability, proposing that the main mechanism of Au decomposition is oxidation of ligands and these could be avoided due to the antioxidant activity of the polyimine. In addition strong Coulombic forces maintain the Au cluster attached to the branched polymer onto the metal oxide surface, avoiding metal leaching. The authors have shown the advantage of the proposed methodology by using the Au cluster on silica coated and coated it with TiO₂, the resulting multicomponent nanoparticulated material acting as visible light photocatalyst for degradation of dyes and pollutants in water. Considering the interest in the use of Au clusters in catalysis and photocatalysis, the proposed methodology could be of large interest. Publication in Nature Communications is recommended with some changes addressing the following comments:

Author reply:

Thank you very much for your comment that this work could be of large interest in the use of Au GSH clusters in catalysis and photocatalysis. We have revised our manuscript with great attention according to your valuable comments.

Comments 1: *The presence of the TiO₂ shell increases considerably adsorption of the dye compared to silica. It is unclear how the authors have taken into account the increased adsorption when obtaining the corrected photocatalytic degradation data.*

Author reply:

Thanks for your valuable comments on our manuscript. To address the reviewer's concern, the Langmuir-Hinshelwood (L-H) kinetic model, which has been widely applied to various heterogeneous photocatalysis systems toward photodegradation of dyes (**Appl. Catal. B** 2004, 49, 1-14; **J. Hazard. Mater.** 2004, 112, 269-278; **CrystEngComm** 2011, 13, 1939-1945; **Phys. Chem. Chem. Phys.** 2012, 14, 82-85; **Mater. Lett.** 2013, 91, 170-174; **Chem. Eng. J.** 2011, 172, 746-753), has been employed to evaluate the reaction rate constant. According to the L-H model, the rate equation can be written as follows:

$$\ln\left(\frac{C_0}{C}\right) + K(C_0 - C) = k_r K t \quad (\text{a})$$

where C_0 is the initial concentration of the reactant, C is the concentration of the reactant, t is the time, K is the adsorption coefficient of the reactant, k_r is the reaction rate constant. Generally, due to the initial concentration of pollutant is low, the equation can be simplified as (**Appl. Catal. B** 2004, 49, 1-14; **J. Hazard. Mater.** 2004, 112, 269-278):

$$\ln\left(\frac{C_0}{C}\right) = k' t \quad (\text{b})$$

where k' is the apparent rate constant. The results of calculated reaction rate constants are

presented in Supplementary Table 2 in the revised supplementary information. It is clearly observed that the reaction rates of various SABT samples increase with increasing thickness of TiO₂ layer and the SABT-0.15 composite exhibits highest reaction rate of 0.119 min⁻¹ due to the presence of optimum TiO₂ shell thickness, which is consistent with the photocatalytic degradation efficiency, as displayed in Fig. 3a.

Additionally, we would like to emphasize that, **the performance of photocatalysts is determined not only by the adsorption ability of catalyst but also the charge carrier separation efficiency.** As shown in Fig. 3g, even though the SABT-0.2 composites have the strongest adsorption toward dyes, the photocatalytic efficiency of SABT-0.2 is lower than that of SABT-0.15 (Fig. 3a), which is attributed to the fact that the dye adsorption and charge carrier separation should have a synergistic effect on the catalytic performance.

Supplementary Table 2. The reaction rate constants for photocatalytic degradation of RhB over different samples.

Samples	SiO ₂	SAB	SABT-0.05	SABT-0.1	SABT-0.15	SABT-0.2	TAB	TiO ₂
Reaction rate (min ⁻¹)	0	0.0143	0.0375	0.0646	0.119	0.0537	0.00645	0.000274

Figure 3 | The enhancement of photocatalytic performance over SABT composites and the underlying mechanism. Photocatalytic degradation of (a) RhB over blank SiO₂ spheres, TiO₂, SAB,

TAB and SABT composites with different TiO₂ shell thickness under visible light irradiation ($\lambda > 420$ nm) for 0.5 h; photocatalytic reduction of (b) *p*-methoxy nitrobenzene to *p*-methoxy aniline and over blank SiO₂ spheres, TiO₂, SAB, TAB and SABT composites with different TiO₂ shell thickness under visible light irradiation for 5 h; (c) recycling photocatalytic degradation of RhB over optimal SABT-0.15 composite under visible light irradiation ($\lambda > 420$ nm); (d) transient photocurrent densities of SiO₂ spheres, SAB and SABT composites with different TiO₂ shell thickness under visible light irradiation ($\lambda > 420$ nm); (e) scheme illustrating the photogenerated electron transport pathways between Au GSH clusters and TiO₂ shell; (f) cyclic voltammograms of SiO₂ spheres, SAB and SABT composites with different TiO₂ shell thickness; (g) bar plots showing the remaining RhB in reaction solutions after being kept in dark for 3 h to achieve the adsorption–desorption equilibrium over SiO₂ spheres, SAB and SABT composites with different TiO₂ shell thickness; (h) nitrogen adsorption–desorption isotherms of SiO₂ spheres, SAB and SABT composites with different TiO₂ shell thickness; (i) surface area of SiO₂ spheres, SAB and SABT composites with different TiO₂ shell thickness.

Corresponding revisions highlighted in red in the revised manuscript:

Lines 2-5 of Page 11:

The pseudo-first order kinetic of the degradation of RhB based on the above data is shown in Supplementary Table 2. It is clear that all of the SABT samples show higher reaction rate than the SAB and TAB samples, among which SABT-0.15 exhibits the highest reaction rate of 0.119 min⁻¹ among these SABT composites.

Corresponding revisions highlighted in red in the revised supplementary information:

Supplementary Table 2 has been added

Comments 2: *It is known that amorphous TiO₂ has much lower catalytic activity than crystalline TiO₂ phases. The present procedure does not allow the typical calcination at 350 °C to crystallize amorphous titania. A comment on this issue (and possible a strategy to circumvent this issue) should be included in the manuscript.*

Author reply:

Thanks for your kind comments. We agree with the reviewer's opinion that the amorphous TiO₂ with defects could exhibit worse photoactivity than crystalline TiO₂, which has also been demonstrated by previous works (*J. Phys. Chem. B* 1997, 101, 3746-3752; *Appl. Catal. B* 2000, 26, 207–215; *J. Phys. Chem. C* 2008, 112, 3083-3089; *J. Mater. Chem.* 2006, 16, 77-82; *Energy Environ. Sci.* 2015, 8, 286-296). According to your precious suggestion, we have paid our endeavors to enhance the crystallization of TiO₂, including calcination and hydrothermal treatment, since the high temperature is generally required to transform amorphous TiO₂ particle into an anatase one.

The sample of SABT-0.15 has been calcinated at 350 °C under Ar for 2 h and the calcinated sample is denoted as SABT-0.15-350. Even though the peak of anatase TiO₂ (A-TiO₂) can be observed in the XRD pattern (Supplementary Fig. 23c), the color of SABT-0.15-350 sample changes to purple, as shown in Supplementary Fig. 23b, which corresponds to the color of Au nanoparticles

(NPs), indicating that the Au GSH clusters aggregate into large Au NPs due to the high calcination temperature. Furthermore, the DRS spectrum of SABA-0.15-350 sample in Supplementary Fig. 23a exhibit a surface plasmon resonance (SPR) peak located at 550 nm belonging to the metallic Au NPs, confirming the fusion of Au GSH clusters into Au NPs with large size. More direct evidence comes from the TEM image of SABA-0.15-350 sample, as pictured in Supplementary Fig. 23d, and the size of Au NPs is demonstrated to be 7.8 nm (Supplementary Fig. 23e), which suggests that the calcination is unsuitable for enhancing the crystallinity of TiO₂ since the high temperature can lead to the aggregation of Au GSH clusters.

We then treat the SABA-0.15 composite under hydrothermal condition at 180 °C for 12 h for crystallizing TiO₂, and the obtained sample is labeled as SABA-0.15-180. The application of elevated temperatures and pressures in an aqueous solution could facilitate the conversion of amorphous TiO₂ into crystalline TiO₂ and cause an increase in its crystallinity. The XRD result in Supplementary Fig. 24c indicates that the TiO₂ in SABA-0.15-180 sample is anatase (A-TiO₂). However, the intelligently designed core-shell structure of SABA-0.15 is destroyed during the hydrothermal process, as revealed by TEM image in Supplementary Fig. 24d. Moreover, the coalescence of Au GSH clusters into larger metallic Au NPs has also been observed and confirmed by a series of techniques, as displayed in Supplementary Fig. 24a, b and d. The size of Au NPs in SABA-0.15-180 sample is calculated to be 3.0 nm (Supplementary Fig. 24e). The fusion of Au GSH clusters may be ascribed to the high temperatures and pressures during the hydrothermal process, which is essential for crystallizing TiO₂.

The photocatalytic performances of SABA-0.15-350 and SABA-0.15-180 composites have been evaluated toward photocatalytic RhB degradation and reduction of *p*-methoxy nitrobenzene under visible light illumination ($\lambda > 420$ nm), as shown in Supplementary Fig. 25. Both the SABA-0.15-350 and SABA-0.15-180 samples exhibit poor photoactivity as compared with SABA-0.15 composite, which could be attributed to the aggregation of Au GSH clusters into large metallic Au NPs and the destroy of core-shell structure of SABA-0.15-180 composite. According to your precious suggestion, these results have been supplemented in our revised manuscript to provide more information for the readership with citation of relevant references in our revised manuscript.

Supplementary Figure 23. (a) DRS spectrum; (b) digital photograph; (c) XRD pattern; (d) HRTEM image of SBT-0.15 composite calcinated at 350 °C under Ar for 2 h (denoted as SBT-0.15-350); (e) size distribution histogram of Au nanoparticles over SBT-0.15-350 sample; (f) EDX spectrum of SBT-0.15-350 originated from **Supplementary Fig. 23d**.

Supplementary Figure 24. (a) DRS spectrum; (b) digital photograph; (c) XRD pattern; (d) HRTEM image of SBT-0.15 composite calcinated at 180 °C for 12 h (denoted as SBT-0.15-180); (e) size distribution histogram of Au nanoparticles over SBT-0.15-180 sample; (f) EDX spectrum of

SABT-0.15-180 originated from **Supplementary Fig. 24d**.

Supplementary Figure 25. Photocatalytic degradation of (a) RhB over SABT-0.15, SABT-0.15-350 and SABT-0.15-180 composites under visible light irradiation ($\lambda > 420$ nm) for 0.5 h; photocatalytic reduction of (b) *p*-methoxy nitrobenzene to *p*-methoxy aniline over SABT-0.15, SABT-0.15-350 and SABT-0.15-180 composites under visible light irradiation ($\lambda > 420$ nm) for 5 h.

Corresponding revisions highlighted in red in the revised manuscript:

Lines 30-35 of Page 11 and Lines 1-4 of Page 12:

In addition to the TiO₂ shell thickness, the crystallization of TiO₂ layer in composites is also a factor that affects the photocatalytic performance of SABT composites and it has been reported that the amorphous TiO₂ usually exhibits poor photocatalytic activity⁵⁸⁻⁶². We have paid our endeavors to crystallize TiO₂ shell in SABT-0.15, including calcination and hydrothermal treatment, since the high temperature is generally required to transform amorphous TiO₂ particle into an anatase one. Even though the crystallization of TiO₂ can be improved by calcination and hydrothermal treatment, the Au GSH clusters in SABT-0.15 suffer from serious instability under high temperature and aggregate into metallic Au nanoparticles with large size, as displayed in Supplementary Fig. 23-24, thereby deteriorating the photocatalytic performance of SABT-0.15 samples toward various photocatalytic reactions under visible light illumination ($\lambda > 420$ nm) (Supplementary Fig. 25).

Page 23, References

Ref. 58-62 have been added.

58. Sakatani Y, Grosso D, Nicole L, Boissiere C, de A. A. Soler-Illia GJ, Sanchez C. Optimised photocatalytic activity of grid-like mesoporous TiO₂ films: effect of crystallinity, pore size distribution, and pore accessibility. *J. Mater. Chem.* **16**, 77-82 (2006).
59. Liu H, Joo JB, Dahl M, Fu L, Zeng Z, Yin Y. Crystallinity control of TiO₂ hollow shells through resin-protected calcination for enhanced photocatalytic activity. *Energy Environ. Sci.* **8**, 286-296 (2015).
60. Zhang Q, Gao L, Guo J. Effects of calcination on the photocatalytic properties of nanosized TiO₂ powders prepared by TiCl₄ hydrolysis. *Appl. Catal., B* **26**, 207-215 (2000).
61. Ohtani B, Ogawa Y, Nishimoto S-i. Photocatalytic activity of amorphous-anatase mixture of

titanium(IV) oxide particles suspended in aqueous solutions. *J. Phys. Chem. B* **101**, 3746-3752 (1997).

62. Tian G, Fu H, Jing L, Xin B, Pan K. Preparation and characterization of stable biphasic TiO₂ photocatalyst with high crystallinity, large surface area, and enhanced photoactivity. *J. Phys. Chem. C* **112**, 3083-3089 (2008).

Corresponding revisions highlighted in red in the revised supplementary information:

Supplementary Figures 22-24 have been added and the corresponding discussions have also been supplemented in the revised Supplementary Information.

Comments 3: *To support the role of polyimine as stabilizer, the results of a control in which the SAP material is irradiated in the presence of an antioxidant will be important.*

Author reply:

Thanks for your valuable comments. To address the reviewer's concern, the sample of SAP has been modified by BPEI (BPEI-SAP) and the as-obtained BPEI-SAP composite is irradiated under visible light ($\lambda > 420$ nm) for 10 h. As displayed in Supplementary Fig. 10a, the HRTEM image of BPEI-SAP after light illumination suggests that the aggregation of Au GSH clusters could be inhibited to some extent. The size of Au GSH clusters in BPEI-SAP after 10 h light irradiation is calculated to be 2.0 nm (Supplementary Fig. 10b), which is smaller than that of SAP after light irradiation (6 nm, Fig. 1f), confirming the critical role of BPEI layer in stabilizing the ultrasmall Au GSH clusters.

Supplementary Figure 10. (a) HRTEM image and (b) size distribution histogram of Au GSH clusters over BPEI modified SAP composite after visible light irradiation ($\lambda > 420$ nm) for 10 h.

Figure 1 | Schematic illustration of the photostability test and characterizations. (a) Schematic illustration of synthesis procedure for SiO₂-Au GSH clusters-BPEI composites (SAB) and photostability testing of as-prepared SAB; transmission electron microscopy (TEM) images of SAB (b) before and (c) after visible light irradiation ($\lambda > 420$ nm) for 10 h and SiO₂-Au GSH clusters-pH (SAP) (e) before and (f) after visible light irradiation for 10 h; the corresponding models of (d) SAB and (g) SAP after visible light irradiation for 10 h; high-resolution X-ray photoelectron spectroscopy (XPS) spectra of (h) S 2p and (i) Au 4f for SAB before/after visible light irradiation; (j) UV-vis diffuse reflectance spectrum (DRS) spectrum of SAP after 10 h visible light irradiation. Insets in the b and c are the corresponding models of SAB and SAP before visible light irradiation. **Note:** The histograms in c and f correspond to the particle size distributions of Au GSH clusters in the SAB and SAP after visible light irradiation for 10 h, respectively. The yellow spheres in Fig.1d represent Au GSH clusters and the purple spheres in Fig. 1g represent Au nanoparticles.

Corresponding revisions highlighted in red in the revised manuscript:

Lines 31-35 of Page 6 and Lines 1-2 of Page 7:

To further confirm the role of BPEI in inhibiting the growth of Au GSH clusters, the SAP sample is subsequently modified by BPEI layer (BPEI-SAP) and the obtained BPEI-SAP composite is irradiated under visible light ($\lambda > 420$ nm) for 10 h. The HRTEM image of BPEI-SAP after light illumination in Supplementary Fig. 10a suggests that the aggregation Au GSH clusters could be prevented to some extent and the size of Au GSH clusters is calculated to be 2.0 nm (Supplementary Fig. 10b), confirming the critical role of BPEI layer in stabilizing the ultrasmall Au GSH clusters.

Corresponding revisions highlighted in red in the revised supplementary information:

Supplementary Figures 10 has been added.

Comments 4: *According to the author's proposal polyimine would undergo oxidation in the way to stabilize the Au cluster and would undergo presumably decomposition? Am I right? This indicates that the branched polyimine will be a sacrificial agent that eventually could lose activity. The authors should comment on this.*

Author reply:

Thank you very much for your valuable comments. We certainly agree with the reviewer that BPEI may be consumed ultimately and the Au GSH clusters over SAB may lead to aggregation after BPEI layer is depleted if we irradiate the SAB sample for a very long time. In our original manuscript, the SAB sample has been irradiated under visible light irradiation ($\lambda > 420$ nm) for 24 h to investigate the efficiency of BPEI on inhibiting the fusion of Au GSH clusters. The results in Supplementary Fig. 8 suggest that the size of Au GSH clusters over SAB after 24 h visible light irradiation maintains unchanged, indicating that the BPEI layer as interfacial modification in the SAB system can provide a long time protection (24 h) with regard to the stabilization of Au GSH clusters under continuous visible light illumination ($\lambda > 420$ nm). **To address the reviewer's concern, the irradiation time over SAB has been further prolonged to 36 h and 48 h and the size information of the Au GSH clusters in SAB has also been studied.** As illustrated in Supplementary Fig. 9 in the revised supplementary information, after 36 h and 48 h visible light irradiation, the size of Au GSH clusters will increase to 2.0 nm and 2.1 nm, respectively. The slight increase of Au GSH clusters size may be attributed to the partial depletion of BPEI layer since the BPEI that serves as a reducing agent would undergo oxidation and/or decomposition in the way to stabilize the Au GSH clusters.

Particularly, we would like to emphasize that, despite various studies have observed the transformation of ultrasmall Au GSH clusters to larger Au NPs (**J. Am. Chem. Soc.** 2014, 136, 6075; **J. Phys. Chem. Lett.** 2013, 4, 2847; **ACS Appl. Mater. Interfaces** 2015, 7, 28105; **Sci. Rep.** 2016, 6, 22742), **the effective control of Au GSH clusters with long-term stability on the substrates under *in situ* photo-irradiation conditions still remains a challenge, which becomes the main bottleneck for the development of Au clusters-based catalysts systems.** Our present work employs the BPEI as interfacial modification of SiO₂-Au GSH clusters composites, which can maintain the size and structure of Au GSH clusters over 24 h under continuous visible light

irradiation ($\lambda > 420$ nm) on the surface of SiO₂ supports. This is the first research work regarding how to stabilize the ultrasmall Au GSH clusters for designing efficient and stable Au GSH clusters-based composite photocatalysts. Additionally, the surface of SAB composites has been subsequently coated by a thickness tunable TiO₂ shell for constructing core-shell SiO₂-Au GSH clusters-BPEI@TiO₂ (SABT) structures, which not only further contributes to stabilizing the ultrasmall Au GSH clusters but also improves the photocatalytic activities of Au GSH clusters during catalytic reactions under visible light illumination. Thus, our joint strategy *via interfacial modification and sequential coating of semiconductor shell* provides a simple and effective approach for stabilizing Au clusters with improved photocatalytic performance.

Supplementary Figure 8. TEM image (a) and HRTEM image (b) of SiO₂-Au GSH clusters-BPEI composites (SAB) after visible light irradiation ($\lambda > 420$ nm) for 24 h; size distribution histogram (c) of Au GSH clusters over SAB after visible light irradiation ($\lambda > 420$ nm) for 24 h.

Supplementary Figure 9. HRTEM images of SAB after visible light irradiation ($\lambda > 420$ nm) for (a) 36 h and (c) 48 h; size distribution histogram (c) of Au GSH clusters over SAB after visible light irradiation ($\lambda > 420$ nm) for (b) 36 h and (d) 48 h; (e) EDX spectrum of SAB originated from Supplementary Fig. 9c.

Note: The EDX spectrum in **Supplementary Fig. 9e** evidences the presence of Au, O and Si elements over SAB sample and the detected element Cu can be attributed to the use of Cu grid, which serves as the support for TEM analysis.

Corresponding revisions highlighted in red in the revised manuscript:

Lines 27-31 of Page 6:

When the irradiation time over SAB is further prolonged to 36 h and 48 h, the size of Au GSH clusters will increase to 2.0 nm and 2.1 nm, respectively, as illustrated in Supplementary Fig. 9. The slight increase of Au GSH clusters size is reasonable, which may be attributed to the partial depletion of BPEI since the BPEI layer that serves as a reducing agent would undergo oxidation and/or decomposition in the way to stabilize the Au GSH clusters.

Corresponding revisions highlighted in red in the revised supplementary information:

Supplementary Figure 9 has been added in the revised supplementary information.

Comments 5: *Similarly, the role of silica on the photocatalytic activity is unclear. Since the authors have shown that the use of branched polyimine serves for different metal oxides, would it be possible to use this approach to deposit Au clusters on anatase TiO₂ and test the photocatalytic activity?*

Author reply:

Thanks for your comments. The insulation SiO₂ spheres has been chosen as the inert supports to predominantly focus on investigating the photosensitizer role of Au GSH clusters in photocatalytic applications due to its rather low optical absorption (Supplementary Fig. 3). Additionally, the SiO₂ spheres with well-defined structure is favorable for the construction of core-shell SiO₂-Au GSH clusters-BPEI@TiO₂ composites with high efficiency since the reported Au GSH clusters-semiconductor composites in literatures are generally fabricated by randomly loading the Au GSH clusters onto the surface of semiconductors (**J. Am. Chem. Soc.** 2014, 136, 6075-6082; **J. Phys. Chem. Lett.** 2013, 4, 2847-2852; **ACS Appl. Mater. Interfaces** 2015, 7, 28105-28109; **Sci. Rep.** 2016, 6, 22742).

To address the reviewer's concern, the Au GSH clusters have been deposited on the surface of anatase TiO₂ (Fig. R1g, as shown below) via a pH value adjusted process and interfacial modification process. The stability of obtained anatase TiO₂-Au GSH clusters-pH composites (ATAP) and anatase TiO₂-Au GSH clusters-BPEI composites (ATAB) have been investigated under visible light illumination ($\lambda > 420$ nm). The HRTEM image in Fig. R1a reveals that the Au GSH clusters in ATAP suffer from fusion even after 0.5 h light irradiation, and the size of Au nanoparticles (NPs) is demonstrated to be 1.9 nm (Fig. R1d). As for the sample of ATAB, after 3 h visible light irradiation, the size and structure of Au GSH clusters remain unchanged (Fig. R1b and e), indicating the important role of BPEI in protecting the Au GSH clusters from being oxidized and restraining the growth of Au GSH clusters. The EDX spectrum in Fig. R1h evidences the presence of Au, O and Ti elements over ATAB sample after light illuminated for 3 h. Unfortunately, when the irradiation time

of ATAB is further extended to 5 h, the Au NPs with size of 1.8 nm are detectable, which confirms the slight aggregation of Au GSH clusters, as shown in Fig. R1c and f.

The fusion of Au GSH clusters on the surface of anatase TiO₂ may be ascribed to the presence of abundance surface hydroxyl group (*Environ. Sci. Technol.* 2013, 47, 2777–2783; *J. Am. Chem. Soc.* 2017, 139, 10020–10028), which could facilitate formation of ·OH that decompose BPEI layer, thus resulting in the formation of Au NPs with large size. The above inference has been evidenced by the synthesis of rutile TiO₂-Au GSH clusters-BPEI composites (RTAB), among which the rutile TiO₂ (Fig. R1i) is obtained by calcinating anatase TiO₂ at 850 °C for 5 h. The sample of RTAB has been exposed to continuous visible light irradiation ($\lambda > 420$ nm) for 10 h under ambient conditions and the size information of Au GSH clusters is given by the TEM analysis. As revealed in Fig. R2a and c, the Au GSH clusters maintain the size of 1.4 nm on the surface of rutile TiO₂ for RTAB composites after photo-irradiation. In contrast, the sample of rutile TiO₂-Au GSH clusters-pH (RTAP) has also been fabricated and irradiated under visible light for 10 h, as illustrated in Fig. R2b and d. The Au GSH clusters have aggregated into Au NPs with size of 3 nm, which indicates the effect of BPEI modification on enhancing the stability of Au GSH clusters under visible light illumination.

The photocatalytic performances of rutile TiO₂ nanoparticles and RTAB composites have been studied by degradation of RhB under visible light irradiation ($\lambda > 420$ nm). As shown in Supplementary Fig. 18, the RhB degradation efficiency over RTAB composites is nearly 96%, which is higher than that of rutile TiO₂ (79%). The observed photoactivity enhancement is attributed to the addition of Au GSH clusters, which could generate electron-hole pairs under visible light irradiation to drive the photocatalytic degradation of RhB. Notably, due to the random loading of ultras-small Au GSH clusters onto TiO₂ supports, the RTAB exhibits moderate photoactivity enhancement than bare rutile TiO₂, which suggests that the random loading of ultras-small Au GSH clusters onto semiconductors without rational structure design is inefficient to achieve high efficient Au GSH clusters-semiconductor composites.

Figure R1. HRTEM image of (a) anatase TiO₂-Au GSH clusters-pH composites (ATAP) after visible light irradiation ($\lambda > 420$ nm) for 0.5 h; HRTEM images of anatase TiO₂-Au GSH clusters-BPEI composites (ATAB) after visible light irradiation ($\lambda > 420$ nm) for (b) 3 h and (c) 5 h; size distribution histogram of Au GSH clusters over (d) ATAP after visible light irradiation ($\lambda > 420$ nm) for 0.5 h; size distribution histograms of Au GSH clusters over ATAB after visible light irradiation ($\lambda > 420$ nm) for (e) 3 h and (f) 5 h; XRD pattern of (g) anatase TiO₂; (h) EDX spectrum of ATAB after visible light irradiation ($\lambda > 420$ nm) for 3 h; XRD pattern of (i) rutile TiO₂.

Figure R2. HRTEM images of (a) rutile TiO₂-Au GSH clusters-BPEI composites (RTAB) and (b) rutile TiO₂-Au GSH clusters-pH composites (RTAP) after visible light irradiation ($\lambda > 420$ nm) for 10 h; size distribution histograms of Au GSH clusters over (c) RTAB and (d) RTAP after visible light irradiation ($\lambda > 420$ nm) for 10 h.

Supplementary Figure 18. Photodegradation of RhB over (a) bare ZnO and ZnO-Au GSH clusters-BPEI composites (ZAB) for 9 h and (b) rutile TiO₂ and rutile TiO₂-Au GSH clusters-BPEI composites (RTAB) for 6 h under visible light irradiation ($\lambda > 420$ nm).

Corresponding revisions highlighted in red in the revised manuscript:

Lines 5-14 of Page 7:

Additionally, the surfaces of different metal oxide supports, including rutile TiO₂, ZnO and ZrO₂, have been positively charged by the BPEI modification (Supplementary Fig. 11), which can subsequently interact with the negatively charged Au GSH clusters (Supplementary Fig. 4c) *via* an

electrostatic self-assembly method to produce metal oxide-Au GSH clusters-BPEI composites (MABs) and the photostability of MABs has been investigated under the same conditions as that for SAB and SAP. As illustrated in Supplementary Fig. 12, the mean diameter of Au GSH clusters on the surfaces of various metal oxide supports after continuous visible light ($\lambda > 420$ nm) for 10 h is determined to be 1.4 nm, indicating the critical role of BPEI on inhibiting the aggregation of Au GSH clusters and excluding the supports effect on the photostability enhancement of Au GSH clusters.

Lines 12-22 of Page 9:

Notably, it has been demonstrated that the CB edge potential of metal oxides (*e.g.*, ZnO and rutile TiO₂) is more positive than the LUMO potential of Au GSH clusters, which enables the transformation of photoexcited electrons from Au GSH clusters to the metal oxide supports^{15, 24, 53}. The photoactivities of BPEI modified metal oxides-Au GSH clusters composites (denoted as MAB) have been evaluated toward RhB degradation under visible light irradiation ($\lambda > 420$ nm). As shown in Supplementary Fig. 18, the samples of MAB exhibit moderate photoactivity enhancement than bare semiconductors, which suggests that the random loading of ultras-small Au GSH clusters onto semiconductors without rational structure design is inefficient to achieve high efficient Au GSH clusters-semiconductor composites. Therefore, a thickness tunable TiO₂ shell has been coated onto the surface of SAB composites for designing core-shell SiO₂-Au GSH clusters-BPEI@TiO₂ structures to construct high performance Au GSH clusters-semiconductor composites for solar energy conversion.

Corresponding revisions highlighted in red in the revised supplementary information:

Supplementary Figure 12 has been revised and **Supplementary Figure 18** has been added. The discussion on the stability of Au GSH cluster on anatase TiO₂ has also been supplemented in the form of Appendix and **Supplementary Figure A1** and **A2** have also been added in the revised supplementary information.

Comments 6: *Which is the Au⁺/Au(0) proportion of the Au GSH clusters determined by XPS?*

Author reply:

Thanks for your valuable comments. As you kindly suggested, the Au⁺/Au⁰ proportion of the Au GSH clusters has been calculated based on the XPS result in Fig. 1a and the Au⁰ content is found to constitute ~90% of all Au atoms.

Corresponding revisions highlighted in red in the revised manuscript:

Lines 30-31 of Page 7:

The Au⁰ content determined by XPS is found to constitute ~90% of all Au atoms.

Comments 7: *It is unclear why the authors use the prefix bio to indicate that the polyethylimine is reducible. Please clarify.*

Author reply:

Thanks for your valuable comments. As you kindly suggested, the prefix “bio” has been deleted in our revised manuscript.

Corresponding revisions highlighted in red in the revised manuscript:**Lines 20-25 of Page 5:**

The SiO₂ spheres become positively charged (Supplementary Fig. 4a) after surface modification with branched poly-ethylenimine (BPEI)³⁶, a conjugated reducible dendrimer, which leads to a substantial electrostatic attraction with negatively charged Au GSH clusters (Supplementary Fig. 4c) by coulombic forces, thereby forming SiO₂-Au GSH clusters-BPEI composites (SAB) with strong interfacial interaction between Au GSH clusters and SiO₂ supports

Comments 8: *It is indicated in the text that Au clusters could be used as visible light photosensitizer. A comment on the λ_{max} absorption and other photophysical relevant properties of Au GSH as photosensitizer would be important.*

Author reply:

Thanks for your valuable comments. As you kindly suggested, the photoluminescence (PL) excitation spectrum of Au GSH clusters has been performed (Supplementary Fig. 1c), which shows a maximum at 400 nm and coincides well with the absorption shoulder (~400 nm) observed in the absorption spectrum. Additionally, more discussions regarding the absorption and photophysical properties of Au GSH clusters have been supplemented in our revised manuscript, which faithfully confirms that the Au GSH clusters can act as a visible light photosensitizer.

UV-vis absorption spectrum of in Supplementary Fig. 1c suggests that the Au GSH clusters show an absorption onset at ~520 nm with a distinct shoulder around 400 nm, which is attributed to the highest occupied molecular orbital-lowest unoccupied molecular orbital (HOMO-LUMO) transition originated from the ligand-to-metal charge transfer, indicating that Au GSH clusters could be used as visible light photosensitizers. The photoluminescence (PL) excitation spectrum exhibits a maximum at 400 nm, which coincides well with the absorption shoulder observed in the absorption spectrum. The PL emissive spectra of Au GSH clusters with different excitation wavelength in Supplementary Fig. 1f show a low energy emission band with the peak maximum at 605 nm, which is ascribed to the triplet metal-centered state, and the shape of the PL spectra is independent of the excitation wavelength. The large Stokes shift in the emission where the absorption band shoulder appears at around 400 nm and the emission maximum is seen at 605 nm, is consistent with the excited state being a ligand-metal charge transfer type.

Supplementary Figure 1. (a) TEM image and (b) size distribution histogram of Au GSH clusters; (c) UV-vis absorption (red), excitation spectrum (green) and emission spectrum (black) of Au GSH clusters in aqueous suspension; digital photographs of Au GSH clusters aqueous solution under the (d) daylight and (e) blacklight illumination; (f) emission spectra of Au GSH clusters aqueous solution under different excitation wavelengths from 380 nm to 500 nm. The insets of a are the model illustrations of Au GSH clusters.

Corresponding revisions highlighted in red in the revised manuscript:

Lines 34-35 of Page 4 and Lines 1-10 of Page 5:

UV-vis absorption spectrum of in Supplementary Fig. 1c suggests that the Au GSH clusters show an absorption onset at ~ 520 nm with a distinct shoulder around 400 nm, which is attributed to the highest occupied molecular orbital-lowest unoccupied molecular orbital (HOMO-LUMO) transition originated from the ligand-to-metal charge transfer,¹⁵ indicating that Au GSH clusters could be used as visible light photosensitizers. The photoluminescence (PL) excitation spectrum exhibits a maximum at 400 nm, which coincides well with the absorption shoulder observed in the absorption spectrum. The PL emissive spectra of Au GSH clusters with different excitation wavelength in Supplementary Fig. 1f show a low energy emission band with the peak maximum at 605 nm, which is ascribed to the triplet metal-centered state³², and the shape of the PL spectra is independent of the excitation wavelength¹⁵. The large Stokes shift in the emission where the absorption band shoulder appears at around 400 nm and the emission maximum is seen at 605 nm, is consistent with the excited state being a ligand-metal charge transfer^{15, 33}.

Page 20, References

Ref. 33 has been added.

33. Wu Z, Jin R. On the ligand's role in the fluorescence of gold nanoclusters. *Nano Lett.* **10**, 2568-2573 (2010).

Corresponding revisions highlighted in red in the revised supplementary information:

Supplementary Figure 1 has been revised.

Comments 9: *Once these points have addressed, publication in Nature Communications should proceed.*

Author reply:

Thanks for your positive and valuable comments.

Thank you very much for your positive, valuable and constructive comments to help us improve the quality of the manuscript with great attention!

Reviewers' Comments:

Reviewer #1 (Remarks to the Author):

Most of the concerns are clarified in the revised version. No more issues with the experimental results.

Reviewer #2 (Remarks to the Author):

Authors have addressed all remarks well, and paper is ready for publication.

Reviewer #3 (Remarks to the Author):

The authors have addressed satisfactorily most of my previous comments and have performed additional experiments to complement the existing data. It appears that the stability of the Au clusters derives from the stability of the glutathione ligands against the oxidative degradation provided by the branched dendritic polyimine polymer. Depletion of this polymer over extended irradiation periods results in aggregation of the clusters and formation of Au nanoparticles. The role of SiO₂ has also been clarified and it has been shown that other metal oxides, particularly TiO₂ rutile can be also suitable, although the gain in efficiency is less notable than in the case of silica. After reading the revision, my previous recommendation for publication in Nature Communications is reinforced, considering the interest in the better use of precious metals, the advantage of use clusters of a few Au atoms and the paucity of studies in this area.

I would ask the authors, however, a last effort trying to perform an additional experiment in which the system SAP is irradiated in the presence of an aqueous antioxidant (like vitamin C) in the absence of the BPEI to determine if the Au cluster stability effect can be also achieved using a soluble antioxidant. In this way, by replenishing this agent it could be possible to stabilize the Au clusters indefinitely.

A last comment is that I do not agree that growing from 1.2 to 2.0 or 2.4 nm is a small particle size increase. What is important is the relative particle size increase and this change is substantial indicating a change from a cluster to nanoparticles.

Publication is recommended.

Point-by-Point Response to Reviewers' Comments

Response to the comment of reviewer 1:

Comment: *Most of the concerns are clarified in the revised version. No more issues with the experimental results.*

Author reply:

We sincerely thank the reviewer for the final evaluation on our manuscript.

Response to the comment of reviewer 2:

Comment: *Authors have addressed all remarks well, and paper is ready for publication.*

Author reply:

We sincerely thank the reviewer for the final evaluation on our manuscript.

Response to the comment of reviewer 3:

Comments:

The authors have addressed satisfactorily most of my previous comments and have performed additional experiments to complement the existing data. It appears that the stability of the Au clusters derives from the stability of the glutathione ligands against the oxidative degradation provided by the branched dendritic polyimine polymer. Depletion of this polymer over extended irradiation periods results in aggregation of the clusters and formation of Au nanoparticles. The role of SiO₂ has also been clarified and it has been shown that other metal oxides, particularly TiO₂ rutile can be also suitable, although the gain in efficiency is less notable than in the case of silica. After reading the revision, my previous recommendation for publication in Nature Communications is reinforced, considering the interest in the better use of precious metals, the advantage of use clusters of a few Au atoms and the paucity of studies in this area.

Author reply:

We do appreciate your positive assessment and helpful comments in the reviewing process, which help us significantly improve the quality of our manuscript.

Comment 1: *I would ask the authors, however, a last effort trying to perform an additional experiment in which the system SAP is irradiated in the presence of an aqueous antioxidant (like vitamin C) in the absence of the BPEI to determine if the Au cluster stability effect can be also achieved using a soluble antioxidant. In this way, by replenishing this agent it could be possible to stabilize the Au clusters indefinitely.*

Author reply:

Thanks for your valuable comments. As you kindly suggested, the SAP sample has been irradiated under visible light for 10 h in the ascorbic acid solution with concentration of 0.1 M. TEM images of SAP after light irradiation (Figure R3) suggest that the size of Au GSH clusters slightly increase from 1.4 nm to 1.6 nm, which indicates that Au GSH cluster suffer from coalescence in the presence of a soluble antioxidant. This can be attributed to the fact that the fusion of Au GSH cluster is not only the oxidation of ligands in the Au GSH cluster but also the subsequent migration of these

clusters to grow into large Au nanoparticles. As compared with the soluble antioxidant ascorbic acid, the BPEI with the multiple polymer chains of primary, secondary and tertiary amine groups can encapsulate the as-synthesized Au GSH clusters *via* crosslinking, which can fix these clusters on the surface of supports and hamper the migration of clusters, thereby stabilizing the ultrasmall Au GSH clusters.

Figure R3. (a) HRTEM image and (b) size distribution histogram of Au GSH clusters over SAP composite after visible light irradiation ($\lambda > 420$ nm) for 10 h in the ascorbic acid solution with concentration of 0.1 M.

Comment 2: A last comment is that I do not agree that growing from 1.2 to 2.0 or 2.4 nm is a small particle size increase. What is important is the relative particle size increase and this change is substantial indicating a change from a cluster to nanoparticles.

Author reply:

Thanks for your valuable comments. We agree with the reviewer that the increase of particle size suggests the change from clusters to nanoparticles. As you kindly suggested, we have revised our manuscript with great attention.

Corresponding revisions highlighted in red in the revised manuscript:

Lines 27-35 of Page 6 and Lines 1-2 of Page 7:

When the irradiation time over SAB is further prolonged to 36 h and 48 h, the size of Au GSH clusters will increase to 2.0 nm and 2.1 nm, respectively, as illustrated in Supplementary Fig. 9, which indicates the slight aggregation of Au GSH clusters. Such size increase of Au GSH clusters may be attributed to the partial depletion of BPEI since the BPEI layer that serves as a reducing agent would undergo oxidation and/or decomposition in the way to stabilize the Au GSH clusters. To further investigate the role of BPEI in inhibiting the growth of Au GSH clusters, the SAP sample is subsequently modified by BPEI (BPEI-SAP) and the obtained BPEI-SAP composite is irradiated under visible light ($\lambda > 420$ nm) for 10 h. The HRTEM image of BPEI-SAP after light illumination in Supplementary Fig. 10a suggests that, as compared with SAP samples (Fig. 1b-d), the serious aggregation of Au GSH clusters could be prevented to some extent and the size of Au GSH clusters is calculated to be 2.0 nm (Supplementary Fig. 10b).

Comment 3: *Publication is recommended.*

Author reply:

We sincerely thank the reviewer for the final evaluation on our manuscript.

Thank you very much for your positive, valuable and constructive comments to help us improve the quality of the manuscript!